# Enhancing Deep Consistent Graph Metric with Affinity and Alignment for Incremental Social Event Detection using Cross-Layer Attention

**Shraban Kumar Chatterjee**                                            *chatterjee.2@iitj.ac.in*
*Department of Computer Science & Engineering*
*Indian Institute of Technology Jodhpur*

**Shubham Gupta**                                                       *gupta.37@iitj.ac.in*
*Department of Computer Science & Engineering*
*Indian Institute of Technology Jodhpur*

**Suman Kundu**                                                         *suman@dsai.iitm.ac.in*
*Wadhwani School of Data Science & AI*
*Indian Institute of Technology Madras*

**Reviewed on OpenReview:** *https://openreview.net/forum?id=vNJ7mCgDbq*

## Abstract

Existing methods of event detection from social media (i.e., X), for instance, KPGNN, FinEvent, and CLKD, use triplet loss for feature separation. Triplet loss suffers from two notable discrepancies in the latent space: (i) inconsistency in intra-event and inter-event distances, and (ii) an inability to ensure the closeness of messages from the same event across different mini-batches. The present paper proposes two novel loss functions to improve consistency in the latent space. The first loss function guarantees consistent intra-event and inter-event distances by increasing the affinity between intra-event points. On the other hand, the alignment loss enhances the cosine similarity between the feature space and label space, thereby aligning features of the same event class across diverse mini-batches. We provide theoretical justification that the proposed loss ensures discriminative features in the latent space, like CGML, without its costly pairwise or specialised batching. Adding to our loss function, we introduce a new attention module designed to effectively address heterogeneous relations without necessitating a separate optimisation objective. Through comprehensive experimentation on two publicly available datasets, we have shown an average improvement of 24.05%, 27.23% and 123.69% in NMI, AMI and ARI, respectively, over supervised SOTA event detection methods. Our method also shows improvements over SOTA unsupervised event detection methods across both datasets. These are supported by statistical significance tests. Generalizability of the proposed loss in general clustering problem in graph domain is shown through experiments.

## Introduction

Events in the real world often manifest in the virtual world, in the form of discussions spanning from government policies to socio-economic status, from summer schools to the European League. These discussions open the avenue for grouping messages on social media into meaningful events. The problem of finding events from social media is known as the event detection problem. Event detection can help in public opinion-based policy making (Mao et al., 2024), crisis management (Saini et al., 2024; Zhang et al., 2022b), sentiment analysis (PETRESCU et al., 2024), and risk management Zhang et al. (2022a), among others. Detecting

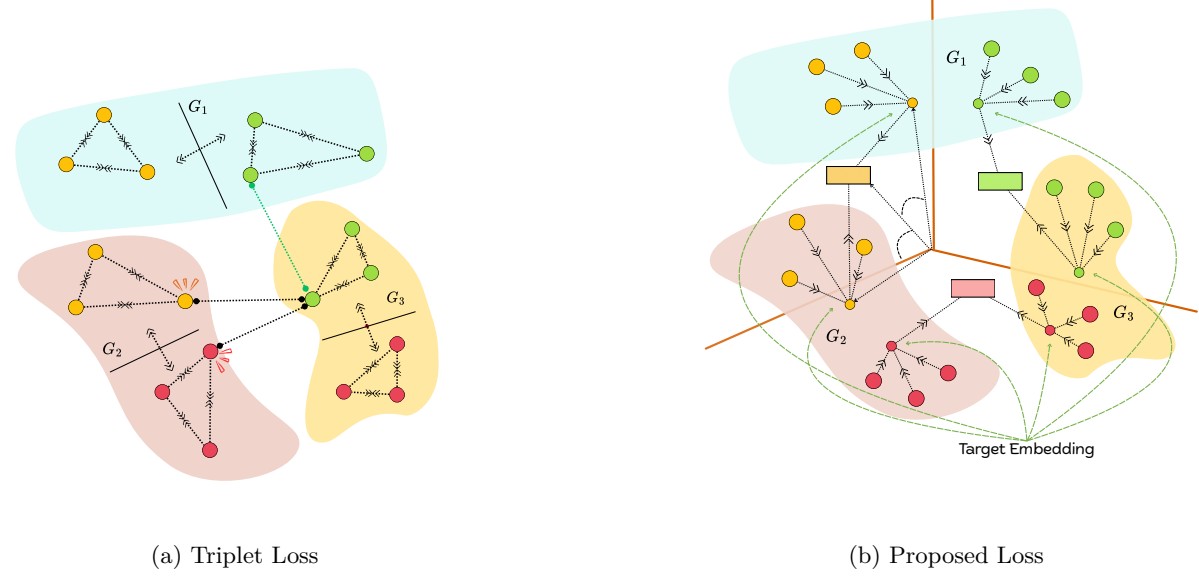

(a) Triplet Loss  (b) Proposed Loss

Figure 1: Comparison between triplet loss and proposed loss. Here, the circles represent the feature space and the rectangles represent the label space.

events from social messages is challenging, specifically due to the large volume of incoming messages that a social event can generate. Another challenge is to determine the relevance of events based on the span of time. As time passes, new events emerge, while older ones fade from discussion. Furthermore, unrelated events often share similar text features, utilising the same sets of words or common hashtags. Finally, social messages contain heterogeneous attributes, including timestamps and user mentions.

In order to address the challenges of scale and event relevance in large message graphs, KPGNN (Cao et al., 2021) partitions the graph into smaller blocks. They employ a triplet loss function to achieve feature separation. However, this approach does not account for heterogeneous attributes within the text. Building upon KPGNN, FinEvent (Peng et al., 2023) incorporates the heterogeneous characteristics by using an agentic framework to select the most salient edges. This method, however, introduces considerable computational overhead. It requires an agent for each heterogeneous relation, and each agent must calculate a reward function based on a neighbourhood sorted by Euclidean distance from the relation graph. Furthermore, the reward function requires a separate optimisation objective, necessitating the generation of ground truth via K-Means for each relation type. For feature generation, it uses the same triplet loss as KPGNN and attempts to mitigate class imbalance by increasing the ratio of negative to positive samples. CLKD (Ren et al., 2024) also extends KPGNN, adapting it for cross-lingual event detection, but it, too, depends on the triplet loss for feature separation. A common characteristic of KPGNN, FinEvent, and CLKD is their fundamental reliance on a triplet loss framework for creating discriminative feature representations. The authors of CGML (Chen et al., 2021) have addressed some limitations of triplet loss by introducing the concept of a discriminative graph. However, to arrive at a discriminative graph, they need a comparison between many graph minibatch

Table 1: Related Work and Baselines

|  | BERT | BiLSTM | EventX | KPGNN | FinEvent | CLKD | HISEvent | HyperSED | Ours |
|---|---|---|---|---|---|---|---|---|---|
| Minibatch Training | ✓ | ✓ | ✗ | ✓ | ✓ | ✓ | ✗ | ✓ | ✓ |
| Relevance | ✗ | ✗ | ✓ | ✓ | ✓ | ✓ | ✓ | ✓ | ✓ |
| Incremental online learning | ✗ | ✗ | ✗ | ✓ | ✓ | ✓ | ✗ | ✗ | ✓ |
| Heterogeneous relations | ✗ | ✗ | ✗ | ✗ | ✓ | ✗ | ✗ | ✗ | ✓ |
| Training on partial Blocks | ✓ | ✗ | ✗ | ✓ | ✓ | ✓ | ✗ | ✗ | ✓ |
| No edge augmentation | ✓ | ✓ | ✓ | ✓ | ✗ | ✓ | ✗ | ✗ | ✓ |
| Decoupled relation objective | – | – | – | – | ✗ | – | – | – | ✓ |
| Supervised/ Semi-supervised | ✗ | ✓ | ✓ | ✓ | ✓ | ✓ | ✗ | ✗ | ✓ |

pairs. This pairwise sampling is costly, especially at high volumes. Also, CGML requires batches with the same number of events in each pair.

We have identified two limitations of the triplet loss as highlighted in Figure (1a). The first is the inconsistency between the intra-event and inter-event distances. Depending on the triplet selection mechanism, in many cases, intra-event feature distance can be comparable to the smallest inter-event feature distance. The same is illustrated in the minibatch $G_2$, where the highlighted yellow and red nodes show inconsistent distances. The second issue is due to the fact that the triplet loss cannot ensure the closeness between nodes of the same event across minibatches. The green line shows the disparity for the green nodes. Some green nodes are closer to yellow (or red) nodes as compared to other green nodes across minibatches.

The present paper proposes two losses, namely, affinity loss and alignment loss, to solve the aforementioned issues with triplet loss by acting as a regularizer. Our losses ensure consistent intra-event and inter-event distances. We achieve this by increasing the affinity between the intra-event points and increasing the cosine similarity between the feature and the label mapped into the same space. We show this by the arrows towards the label space in Figure (1b). This approach aligns the features of the same event within and across different minibatches. Unlike CGML, where pairwise batching and comparison are required, our method works by iterating over the minibatches of a graph, which is linear in the number of minibatches. We theoretically justify that the joint optimisation of the proposed affinity and alignment loss is aligned with the definition of discriminative graphs in CGML. Besides, we introduce a new attention module to extract discriminative features from heterogeneous relations for event detection. The proposed attention module does not require a separate optimisation function like FinEvent.

## Related Work

One of the pioneering works in event detection is EventX (Liu et al., 2020), which uses keyword commonness. It is designed to extract events from long news articles. Its dependence on TF-IDF makes it less sensitive to low-frequency keywords, which are common in social messages. Other notable works that use word commonness in a given time frame are Li et al. (2012); Marcus et al. (2011). Topic modelling-based approaches Xie et al. (2013); Zhou et al. (2015); You et al. (2013) for event detection operate by assigning each tweet a probabilistic distribution over multiple latent topics. These approaches are primarily designed to work in an offline setting.

Other than the supervised methods discussed earlier, recently proposed unsupervised event detection methods like HISEvent (Cao et al., 2024) and HyperSED (Yu et al., 2025) first construct a message graph and then find message combinations that minimise the 2D entropy. HISEvent is time-consuming as it uses a greedy hierarchical optimisation objective, making it difficult to apply on a large scale. Further, HISEvent does not support minibatching. HyperSED introduces the concept with an anchor message graph construction and designs a hyperbolic GNN with an anchor adjacency matrix reconstruction objective. HyperSED requires training on all the blocks and does not share learning across the blocks. Furthermore, these methods still require edge supplementation. All these limit its scalability in event detection. Table 1 shows a comparison of different event detection methods.

**Deep Metric Learning:** Deep metric learning is currently an active area of research. Pair-based formulations are the most popular in metric learning, with works like Triplet Loss (Sohn, 2016), Contrastive loss (Sun et al., 2014), and N-Pair loss (Sohn, 2016). Instead of pairing nodes, CGML (Chen et al., 2021) looks at it from a graph pair perspective. Its optimisation function is based on increasing the feature similarity between graph minibatches. Proxy-based methods, instead of comparing data points to each other, learn a set of proxies, which serve as representatives for each class, and optimise data-to-proxy relationships. A popular work in this line is Gu et al. (2021). Some researchers have focused on improving metric learning by incorporating principles from information retrieval and ranking. A notable work in this line is Fu et al. (2021), where the authors present a self-supervised auxiliary framework for improved metric learning in information retrieval; similarly, Liao et al. (2023) propose a contextual loss that optimizes the semantic consistency of neighbor sets to enhance retrieval ranking.

# Methodology

**Tweet**

The 2nd #PresidentialDebate between President @BarackObama & Governor @MittRomney takes place tonight at 9 PM EST @HofstraU! Check it out!

**Tweets with common words**

- Foreign policy is the presidents strength. **Tonight** he will need to show why he deserves to be Commander-in-Chief #Debate2012 #LetsTalk"
- we had a earthquake in maine **tonight**. crazy
- 2013 budget: Nigeria should expect more Boko Haram insurgency – Former Yobe **governor** warns FG | AmehDaily
- I love how the moderator is putting @MittRomney in his **place** already! #PresidentialDebate

**Tweets with common entities**

- Video: Expectations and opinions for the second **#PresidentialDebate** http://t.co/jzeVRDBZ
- #Debate2012 coming up. @BarackObama get him this time. **#PresidentialDebate** #Debates
- Before the #debate **tonight**, tell candidates enough is enough! WATCH: http://t.co/2xi6ZaQE and SIGN: http://t.co/JN0jMek5 #debate2012

**Tweets with common mentioned users**

- .@MittRomney & @PaulRyanVP you are in our prayers for a huge win on Nov 6. #RomneyRyan2012 #GOP2012 #Debate2012
- Proud 2 see my alma mater, @HofstraU, getting all this national press for hosting its 2nd #PresidentialDebate in 4 yrs! #RollPride #Hofstra

Figure 2: Example of related tweets extracted based on common words, entities, and mentioned users from the Event2012 dataset.

We have divided the tweets into blocks based on time following Cao et al. (2021). From each block $k$, we have created a Heterogeneous Multilayer Graph (HMG) Chatterjee & Kundu (2024) $H_k$. In each $H_k$, there are 4 layers, where each layer represents a relation graph with nodes representing a tweet $s_i$. We take 3 types of relations between tweets, namely, common entities, common words, and common users, to constitute three layers. Two nodes in layer 4 will contain an edge if $\exists$ an edge between those two nodes in any of the other 3 layers. We assume a node is present in all layers. One may note that common and rare words are excluded from tweets while extracting relationships.

## Initial Feature Generation

We extracted text and timestamp embedding Cao et al. (2021) of a tweet and concatenated them for its initial embedding. Two tweets from the same event can arrive at different points in time. Adding timestamp information separates past and present tweets. We will use the term $\mathcal{X}^0$ to represent the initial features generated in this stage.

## Layer Wise Feature Generation

We extract layer-wise features for a node from a GAT Veličković et al. (2018) encoder $\mathcal{E}_\theta(G_k, \mathcal{X}) \to \mathcal{X}_{G_k}; G_k \in H_k$, parameterised by $\theta$. We use GAT as it weighs each incoming connection of a node in a layer that complements the cross-layer attention mechanism (explained later) we design. We define the embeddings generated from the encoder model as:

$$\mathcal{X}_{s_i}^{l-1} = \|_{h=1}^{H} \mathcal{R}(\sum_{s_j \in N_{s_i}} \Upsilon_{s_i s_j}^{l-1} W^h \mathcal{X}_{s_j}^{l-2}),$$

$$\mathcal{X}_{s_i}^{l} = \mathcal{R}(\sum_{h=1}^{H} \sum_{s_j \in N_{s_i}} \Upsilon_{s_i s_j}^{l} W^h \mathcal{X}_{s_j}^{l-1})$$

$$\Upsilon_{s_i s_j}^{l} = \frac{e^{\varphi(W^T[\mathcal{X}_{s_i}^{l-1} || \mathcal{X}_{s_j}^{l-1}])}}{\sum_{\forall \mathcal{N}_{s_n} \in V} e^{\varphi(W^T[\mathcal{X}_{s_i}^{l-1} || \mathcal{X}_{s_n}^{l-1}])}}$$

(1)

Here, $\mathcal{X}_{s_i}^{l-1}$ and $\mathcal{X}_{s_i}^{l}$ represents the encoding of a node $s_i$ in $(l-1)$th and final GAT layer respectively. $H$, $\mathcal{R}$, $\varphi$, ($W^T, W^h \in \theta$), and $\|$ denote the number of attention heads, ReLU, LeakyReLU, learnable weight matrices, and concatenation operator, respectively. $\Upsilon$ represents the attention score between two nodes $s_i$ and $s_j$. $N_{s_i}$ and $\mathcal{X}_{s_i}^0$ is the neighboring nodes and initial feature vector of node $s_i$.

**Multi-head cross-layer attention**

We extract the node features for every layer (of a block) using the encoder $\mathcal{E}$. Data from each layer may not be significant for a node. We can see from the example, shown in Figure 2, that connections based on common words introduce noise. To address this, we designed a multi-head cross-layer attention network. Our proposed network extracts important node features from each layer. We define the cross-attention for layers $G_i$, $G_j$ in a block as follows:

$$\mathcal{A}(Q_{G_1}, K_{G_2}, V_{G_2}) = \mathcal{S}_\sigma\left(\frac{Q_{G_1} K_{G_2}^T}{\sqrt{d}}\right) V_{G_2}$$

$$\mathcal{M}(Q_{G_1}, K_{G_2}, V_{G_2}) = ||(\mathcal{A}_1, \mathcal{A}_2, ..., \mathcal{A}_k)$$

(2)

Here, Q represents the query for the layer $G_i$, K and V are the keys and values for the layer $G_j$. $\mathcal{S}_\sigma$, $\mathcal{M}$, and $d$ are the softmax function, multi-head attention, and node feature dimension, respectively. Node features from the combined layer $G_c$ contain information from all the relations. So, we use the combined layer $G_c$ as the query $G_i$. We set $G_j$ to the desired layer with which we need to find the attention score. Querying on $G_c$ shows how much each layer contributes to the homogeneous layer. Thus, this formulation evaluates the importance of each relation in $G_c$.

First, we find the contribution of common entities ($G_e$) in the feature vector of a node. For this, we query $G_c$ with key and value as $G_e$ as follows:

$$\mathcal{X}_{G_e}^{h_i} = \mathcal{S}_\sigma\left(\frac{[W_Q \mathcal{X}_{G_c}][W_K \mathcal{X}_{G_e}]^T}{\sqrt{d}}\right) W_V \mathcal{X}_{G_e}$$

(3)

Here, $[W_Q \mathcal{X}_{G_c}]$ and $[W_K \mathcal{X}_{G_e}]$ are the transformed query and key for calculating the attention score. In our case, $W_V = W_K$ as we use the same graph for key and query. $\mathcal{X}_{G_e}^{h_i}$ is a single head of the entity contextualized representation. $\mathcal{X}_{G_e}^{final} = \mathcal{M}(\mathcal{X}_{G_e}^{h_i})$ is the final entity contextualized node representation in the layer $G_e$. Similar to this, we measure the importance of common words and common user mentions in the representation of a node as follows:

$$\mathcal{X}_{G_w}^{final} = \mathcal{M}\left(\mathcal{S}_\sigma\left(\frac{[W_Q \mathcal{X}_{G_c}][W_K \mathcal{X}_{G_w}]^T}{\sqrt{d}}\right) W_V \mathcal{X}_{G_w}\right)$$

(4)

$$\mathcal{X}_{G_u}^{final} = \mathcal{M}\left(\mathcal{S}_\sigma\left(\frac{[W_Q \mathcal{X}_{G_c}][W_K \mathcal{X}_{G_u}]^T}{\sqrt{d}}\right) W_V \mathcal{X}_{G_u}\right)$$

(5)

The final node features after the cross-layer attention phase are as follows:

$$\mathcal{X}_{attn} = \left(||(\mathcal{X}_{G_c}, \mathcal{X}_{G_e}^{final}, \mathcal{X}_{G_w}^{final}, \mathcal{X}_{G_u}^{final})\right) W_{attn}^T + b_{attn}$$

(6)

**Proposed Loss**

We first outline the theoretical foundation before defining the losses. We begin by introducing the definition of a Discriminative Graph from CGML.

**Definition 1.** *Given node features $\mathcal{X}$ and $l$ events, we represent maximum intra-event feature distance for the c-th event as $\alpha_c$ and the minimum inter-event feature distance from the centre of the c-th event to the centre of a negative event as $\beta_c$. A complete graph $G$ with edge weight between nodes i and j as $s_{ij} = \exp\left(-\frac{||f(x_i) - f(x_j)||_2^2}{\sigma}\right)$ is called discriminative if it has $\alpha_c < \beta_c \; \forall c \in l$. Here $\sigma$ is the temperature parameter.*

Note that, for the following proofs, we adopted the necessary condition of Proposition 1 in CGML using the same notations. The details of the adoption and corresponding proof are in the supplementary. Here $f$ represents our encoder $\mathcal{E}$. Furthermore, the following derivations are independent of the internal components of encoder $f$.

**Lemma 1.** *Given encoded feature of a node as $f(.)$, and events $C = \{c_1, c_2, ..., c_k\}$, subgraphs $G'$ and $G''$ sampled from a discriminative graph $G$ with adjacency matrices $S'$ and $S''$, both containing $|c_i'| \le |c_i|$ nodes for event $c_i$, $S_{ij}' \approx S_{ij}''$ when $f(x_i) \approx f(x_j)$ for all $x_i$ and $x_j$ in the same event.*

*Proof.* When we consider $x_i$ from $S'$ and $x_j$ from $S''$ we have $\mathbb{E}[(s'_{ij} - s''_{ij})^2] \leq 2\left(1 - \exp(\frac{2\alpha_c^2}{\sigma})\right)$ (from Proposition 1). In our case $\alpha_c = ||f(x_i) - f(x_j)||_2^2$, so when $f(x_i) \to f(x_j), \alpha_c \to 0$ i.e., $\lim_{f(x_i) \to f(x_j)} ||f(x_i) - f(x_j)||_2^2 = 0$. Thus, $\mathbb{E}[(s'_{ij} - s''_{ij})^2] \to 0$ indicating $s'_{ij} \approx s''_{ij}$. $\qquad\square$

**Lemma 2.** *Given encoded feature of a node as $f(.)$, and events $C = \{c_1, c_2, ..., c_k\}$, subgraphs $G'$ and $G''$ sampled from a discriminative graph $G$ with adjacency matrices $S'$ and $S''$, both containing $|c'_i| \leq |c_i|$ nodes for event $c_i$, $S'_{ij} \approx S''_{ij}$ when $||(f(x_i) - f(x'_i)||_2^2) \to 0$ and $(||f(x_i) - f(x_j)||_2^2) \to 1$ where $x_i$ and $x'_i$ belongs to the same event and $x_i$ and $x_j$ belong to different events.*

*Proof.* We take the inequality $|s'_{ij} - s''_{ij}| \leq 2(b-a)\sqrt{\frac{log(\frac{4}{\delta})}{2}}$ (from Proposition 1). Here $a = \exp\left(-\frac{(\alpha_c + \alpha_k + 2\pi)^2}{4\sigma}\right)$ and $b = \exp\left(-\frac{(2\pi - \alpha_c - \alpha_k)^2}{4\sigma}\right)$ where $\pi$ is the maximum inter-class distance between any two events. From the Lemma we have $\alpha_c = ||(f(x_c) - f(x'_c)||_2^2) \to 0$ and $\alpha_k = ||(f(x_k) - f(x'_k)||_2^2) \to 0$ where $(x_c, x'_c)$ and $(x_k, x'_k)$ belong to events $c$ and $k$ respectively making inter-event distance $\pi = (||f(x_i) - f(x_j)||_2^2) \to 1$. Therefore, $\lim_{(\alpha_c, \alpha_k, \pi) \to (0,0,1)} \exp\left(-\frac{(\alpha_c + \alpha_k + 2\pi)^2}{4\sigma}\right) = \lim_{(\alpha_c, \alpha_k, \pi) \to (0,0,1)} \exp\left(-\frac{(2\pi - \alpha_c - \alpha_k)^2}{4\sigma}\right) = \exp\left(-\frac{1}{\sigma}\right)$. From this we can derive $\lim_{b \to a} 2(b-a)\sqrt{\frac{log(\frac{4}{\delta})}{2}} \to 0$. Thus we prove that $s'_{ij} \approx s''_{ij}$. $\qquad\square$

**Lemma 3** (Sufficiency Condition). *Given any independent subgraph $G'$ of $G$ with adjacency matrix $S'$ containing $|c'_i| \leq |c_i|$ nodes $\forall$ event $c_i \in C = \{c_1, c_2, ..., c_k\}$, $G$ is discriminatory if $||(f(x_i) - f(x'_i)||_2^2) \to 0 \ \forall x_i, x_j$ of the same event and $(||f(x_i) - f(x_j)||_2^2) \to 1 \ \forall x_i, x_j$ in different events hold.*

*Proof.* (Proof by contradiction:) Given the conditions of the Lemma let us assume that the graph $G'$ is not discriminatory. This means, $\exists c_i$ such that: $\alpha_i \geq \beta_i$. From $||(f(x_i) - f(x'_i)||_2^2) \to 0$, we have $\alpha_i \to 0, \forall c_i$. From $(||f(x_i) - f(x_j)||_2^2) \to 1$ we have the squared distance between points in different event approaching 1. Since $\alpha_i \to 0$, each class collapses to its center $C_i$. Thus, the inter-class margin $\beta_i$ for an event $c_i$ is bounded by a positive constant $\beta_i = ||C_i - C_k||_2^2 \to 1, \forall c_i, c_k$. Based on our initial assumption this implies $0 \geq 1, \forall c_i$. Thus we reach a contradiction. Therefore, the conditions of the Lemma are sufficient to ensure a Discriminative Graph. $\qquad\square$

**Remark 1** (Any graph G that satisfies Lemma 3 also satisfies the necessary condition of Proposition 1). *There always exists another subgraph $G''$ (with adjacency matrix $S''$) with $C = \{c_1, c_2, ..., c_k\}$ events that can be randomly sampled from $G$, where both $G'$ and $G''$ have $|c'_i| \leq |c_i|$ points in event $c_i$. As $||(f(x_i) - f(x'_i)||_2^2) \to 0$ and $(||f(x_i) - f(x_j)||_2^2) \to 1$, we have from Lemmas 1 and 2, $S' \approx S''$.*

**Lemma 4.** *Let $S', S'',$ and $L$ matrices $\in \mathbb{R}^{m \times n}$. If $||S' - L||_F^2 \to 0$ and $||S'' - L||_F^2 \to 0$, then $||S'' - S'||_F^2 \to 0$.*

*Proof.* Proof in supplementary. $\qquad\square$

**Lemma 5.** *Given encoders $\mathcal{F}_x$ and $\mathcal{F}_y$, we sample (iid) $|c'_i| \leq |c_i|$ nodes from each event $c_i$ two times with feature matrices $X', X'' \in \mathbb{R}^{d*(|c'_i| + \cdots + |c'_k|)}$, and adjacency matrices $S'$ and $S''$ respectively then $|\mathcal{F}_x(S'X'^T) - \mathcal{F}_x(S''X''^T)| \leq |\mathcal{F}_x(S'X'^T) - \mathcal{F}_y(L)| + |\mathcal{F}_x(S''X''^T) - \mathcal{F}_y(L)|$. Here $L$ denotes the label vector where $l_i \in L$ denotes that node $x \in c_i$ has label $l_i$.*

*Proof.* For this proof, we assume that the nodes in $X'$ and $X''$ are arranged in the same order of events. Since both $X'$ and $X''$ have the same number of events with $|c'_i|$ nodes in an event we have $\mathcal{F}_y(L') = \mathcal{F}_y(L'') = \mathcal{F}_y(L)$. Thus, from the inequality $||A - B||_F \leq ||A - C||_F + ||B - C||_F$ (Proof in supplementary) we have:

$$|\mathcal{F}_x(S'X'^T) - \mathcal{F}_x(S''X''^T)| \leq |\mathcal{F}_x(S'X'^T) - \mathcal{F}_y(L)| + |\mathcal{F}_x(S''X''^T) - \mathcal{F}_y(L)|.$$

$\qquad\square$

**Affinity Loss:**

Based on Lemmas 1, 2, and 3, we design Affinity Loss with the aim of taking intra-class target embedding distances close to 0 and inter-class target embedding distances towards 1. This will help us approach a discriminative graph with better event separation. We define our proposed affinity loss function as follows:

$$\mathcal{L}_{affinity} = \|S_{feature} - S_{label}\|_2$$
$$S_{feature}(i,j) = \exp\left(-\frac{\|x_i - x_j\|^2}{\sigma}\right)$$

(7)

$$S_{label}(i,j) = \begin{cases} 0 & \text{if } i = j \text{ (diagonal excluded)} \\ m_1 & \text{if } i \neq j \text{ and } y_i = y_j \text{ (same labels)} \\ m_2 & \text{if } i \neq j \text{ and } y_i \neq y_j \text{ (different labels)} \end{cases}$$

$$\mathcal{L}_{affinity} = \sqrt{\sum_{i=1}^{n}\sum_{j=1}^{n}\left(\exp\left(-\frac{\|x_i - x_j\|^2}{\sigma}\right) - S_{label}(i,j)\right)^2}$$

(8)

When we set $m_1 = 1$ and $m_2 = 0$, $\mathcal{L}_{affinity}$ will be closer to zero for a pair of nodes when embeddings of the same clusters ($S_{label}(ij) = 1$) become closer ($\exp(-\downarrow)$). It will also approach zero when embeddings of different clusters ($S_{label}(ij) = 0$) become farther ($\exp(-\uparrow)$). In the case when embeddings of a node pair in different clusters come close ($S_{label}(ij) = 0, \exp(-\downarrow)$) or embeddings of the nodes in the same cluster become different ($S_{label}(ij) = 0, \exp(-\downarrow)$), the loss value will tend towards one. We use $\uparrow$ and $\downarrow$ to indicate higher and lower values, respectively. The temperature parameter $\sigma$ allows us to adjust the sensitivity, especially in cases where we may not want to penalise inter-cluster distances till a threshold. In our case, we set our temperature value to 1.

**Alignment Loss:**

Based on Lemmas 4 and 5, we define an alignment loss function to encourage alignment between the feature space ($\mathcal{X}_{attn}$) and the label space ($L$). This alignment is measured using cosine similarity between learned features and label embeddings. It is defined as:

$$\mathcal{L}_{\text{align}} = 1 - \frac{1}{n}\sum_{i=1}^{n}\cos(f_x(x_i), f_y(y_i))$$

(9)

The embedding functions $f_x$ and $f_y$ map input features $x_i$ and (integer) label $y_i$ into a shared latent space using two-layer neural networks:

$$f_x(x_i) = W_2^{(x)} \cdot \mathcal{R}(W_1^{(x)}x_i + b_1^{(x)}) + b_2^{(x)}$$

(10)

$$f_y(y_i) = W_2^{(y)} \cdot \mathcal{R}(W_1^{(y)}y_i + b_1^{(y)}) + b_2^{(y)}$$

(11)

Let $e_i^{(x)} = f_x(x_i) \in \mathbb{R}^{d*1}$ and $e_i^{(y)} = f_y(y_i)$. The cosine similarity between the two embeddings is then given by:

$$\cos(f_x(x_i), f_y(y_i)) = \frac{e_i^{(x)\top}e_i^{(y)}}{\|e_i^{(x)}\|_2 \cdot \|e_i^{(y)}\|_2}$$

(12)

Substituting into the loss, we obtain:

$$\mathcal{L}_{\text{align}} = 1 - \frac{1}{n}\sum_{i=1}^{n}\frac{e_i^{(x)\top}e_i^{(y)}}{\|e_i^{(x)}\|_2 \cdot \|e_i^{(y)}\|_2}$$

(13)

This loss function encourages directional similarity between the learned feature and label embeddings for each node. When the embeddings are perfectly aligned (i.e., the angle between them is 0), the cosine similarity is

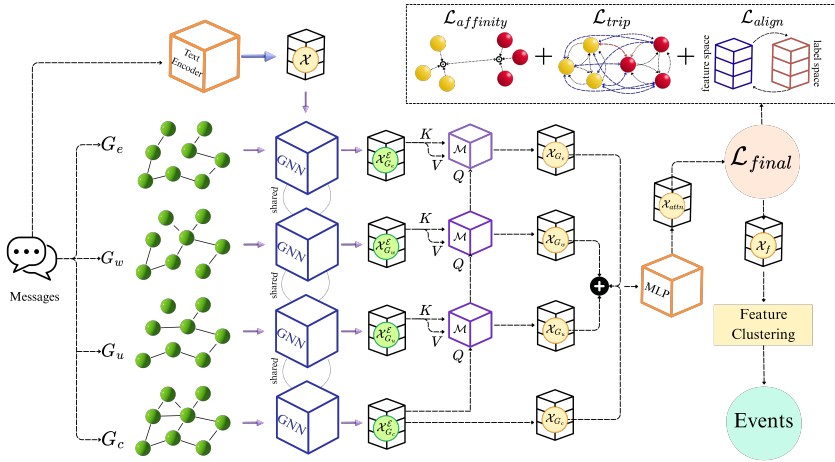

Figure 3: The architecture of the proposed model.

1 and the loss contribution is 0. When the embeddings are orthogonal, the cosine similarity becomes 0, and the loss contribution is 1. In the worst case, when the embeddings point in opposite directions, the cosine similarity becomes -1, resulting in a loss contribution of 2. Note that, a new event class when appear during incremental-training phase is assigned with a new integer number and mapped to the same label space by $f_y$ keeping the projections of old event classes. For both initial training and incremental-training, the projection vector (label embedding) dimensions are kept the same.

In CGML, the authors sample two graphs. Both of them have the same number of nodes, with an equal number of nodes in each class. We decouple the graph regularisation and upper bound of the graph consistency in Eqn. 1 and Eqn. 3 of CGML using our Lemma 4 and Lemma 5 , respectively. This reduces number of samplings that we need for a single epoch making the loss additive as we progress with the batching.

**Incremental Block Wise Training**

For training, we use the proposed affinity and alignment losses as a regularizer to the triplet loss ($\mathcal{L}_{trip}$) in the model. We train our encoder model using the final loss $\mathcal{L}_{affinity} + \mathcal{L}_{align} + \mathcal{L}_{trip}$. This refines the features from the attention step $\mathcal{X}_{attn}$, and we obtain the final features $\mathcal{X}_{final}$. In our incremental setting, the encoder is trained sequentially on data blocks, updating its parameters after each block to retain previously learned information without retraining from scratch. We follow the "latest message" training strategy of Cao et al. (2021) with the loss function defined above. After training, the resulting features can be segregated using any clustering algorithm. This approach also allows for efficient inference, where a model trained on one block can be applied to $w$ subsequent blocks.

**Time Complexity**

For a block $H_k$ with $V$ nodes and $E$ edges across all the layers, the time for constructing or updating the heterogeneous message graphs is $O(E)$. The propagation step requires $O(E)$ time. For a minibatch, the sampling step requires $n_1 * n_2 * ... * n_l$ time, where $n_i$ is the number of neighbours sampled for a node in the GAT layer and $l$ is the number of layers. The propagation step from the GAT module requires $O(Bd_{in}d_{out} + B_e d_{out})$ time, where $d_{in}, d_{out}, B$, and $B_e$ are the input, output dimensions of GAT, node and edge size of the minibatch respectively. The loss functions and attention module require $O(B^2)$ and $O(B^2 d_{out})$ time. The practical running times are shown in the supplementary.

Table 2: Event Detection performance for various traditional and supervised methods in terms of ARI.

| Block | BERT | BiLSTM | EventX | KPGNN | $\text{KPGNN}_t$ | $\text{FinEvent}_k$ | CLKD | Contextual | **Proposed** |
|---|---|---|---|---|---|---|---|---|---|
| | | | | | English | | | | |
| $B_1$ | $0.03 \pm .00$ | $0.15 \pm .00$ | $0.00 \pm .00$ | $0.02 \pm .01$ | $0.03 \pm .02$ | $0.05 \pm .02$ | $0.04 \pm .00$ | $0.04 \pm 0.02$ | $\mathbf{0.97 \pm 0.01}$ |
| $B_2$ | $0.24 \pm .00$ | $0.02 \pm .00$ | $0.42 \pm .00$ | $0.60 \pm .02$ | $0.65 \pm .03$ | $0.68 \pm .01$ | $0.69 \pm .01$ | $0.66 \pm 0.03$ | $\mathbf{0.90 \pm 0.02}$ |
| $B_3$ | $0.11 \pm .00$ | $0.08 \pm .00$ | $0.04 \pm .00$ | $0.44 \pm .03$ | $0.50 \pm .01$ | $0.52 \pm .00$ | $0.56 \pm .02$ | $0.54 \pm 0.04$ | $\mathbf{0.96 \pm 0.03}$ |
| $B_4$ | $0.02 \pm .00$ | $0.01 \pm .00$ | $0.03 \pm .00$ | $0.22 \pm .02$ | $0.24 \pm .02$ | $0.27 \pm .01$ | $0.22 \pm .01$ | $0.29 \pm 0.04$ | $\mathbf{0.87 \pm 0.02}$ |
| $B_5$ | $0.02 \pm .00$ | $0.06 \pm .00$ | $0.09 \pm .00$ | $0.32 \pm .01$ | $0.36 \pm .02$ | $0.49 \pm .00$ | $0.32 \pm .03$ | $0.48 \pm 0.02$ | $\mathbf{0.91 \pm 0.01}$ |
| $B_6$ | $0.02 \pm .00$ | $0.02 \pm .00$ | $0.1 \pm .00$ | $0.46 \pm .01$ | $0.61 \pm .04$ | $0.51 \pm .01$ | $0.62 \pm .01$ | $0.64 \pm 0.04$ | $\mathbf{0.95 \pm 0.01}$ |
| $B_7$ | $-0.02 \pm .00$ | $0.14 \pm .00$ | $0.00 \pm .00$ | $0.06 \pm .01$ | $0.06 \pm .02$ | $0.08 \pm .01$ | $0.07 \pm .01$ | $0.09 \pm 0.03$ | $\mathbf{0.93 \pm 0.01}$ |
| $B_8$ | $0.09 \pm .00$ | $0.02 \pm .00$ | $0.09 \pm .00$ | $0.46 \pm .04$ | $0.50 \pm .01$ | $0.56 \pm .02$ | $0.46 \pm .01$ | $0.51 \pm 0.03$ | $\mathbf{0.88 \pm 0.04}$ |
| $B_9$ | $0.01 \pm .00$ | $0.18 \pm .00$ | $0.06 \pm .00$ | $0.28 \pm .02$ | $0.30 \pm .03$ | $0.39 \pm .01$ | $0.59 \pm .01$ | $0.41 \pm 0.02$ | $\mathbf{0.91 \pm 0.02}$ |
| $B_{10}$ | $0.04 \pm .00$ | $0.01 \pm .00$ | $0.07 \pm .00$ | $0.57 \pm .02$ | $0.50 \pm .01$ | $0.56 \pm .00$ | $0.52 \pm .02$ | $0.60 \pm 0.02$ | $\mathbf{0.95 \pm 0.02}$ |
| $B_{11}$ | $-0.02 \pm .00$ | $0.04 \pm .00$ | $0.04 \pm .00$ | $0.43 \pm .01$ | $0.37 \pm .02$ | $0.34 \pm .01$ | $0.36 \pm .01$ | $0.27 \pm 0.02$ | $\mathbf{0.95 \pm 0.01}$ |
| $B_{12}$ | $0.10 \pm .00$ | $0.03 \pm .00$ | $0.03 \pm .00$ | $0.26 \pm .01$ | $0.32 \pm .03$ | $0.38 \pm .01$ | $0.34 \pm .01$ | $0.28 \pm 0.01$ | $\mathbf{0.78 \pm 0.01}$ |
| $B_{13}$ | $0.01 \pm .00$ | $0.04 \pm .00$ | $0.01 \pm .00$ | $0.30 \pm .02$ | $0.22 \pm .02$ | $0.17 \pm .00$ | $0.33 \pm .00$ | $0.23 \pm 0.03$ | $\mathbf{0.96 \pm 0.02}$ |
| $B_{14}$ | $0.08 \pm .00$ | $0.02 \pm .00$ | $0.03 \pm .00$ | $0.22 \pm .02$ | $0.21 \pm .01$ | $0.35 \pm .01$ | $0.22 \pm .01$ | $0.29 \pm 0.02$ | $\mathbf{0.94 \pm 0.02}$ |
| $B_{15}$ | $0.01 \pm .00$ | $0.03 \pm .00$ | $0.01 \pm .00$ | $0.10 \pm .01$ | $0.07 \pm .03$ | $0.16 \pm .01$ | $0.41 \pm .02$ | $0.09 \pm 0.02$ | $\mathbf{0.92 \pm 0.01}$ |
| $B_{16}$ | $0.00 \pm .00$ | $0.03 \pm .00$ | $0.04 \pm .00$ | $0.44 \pm .01$ | $0.45 \pm .01$ | $0.48 \pm .00$ | $0.57 \pm .02$ | $0.64 \pm 0.02$ | $\mathbf{0.96 \pm 0.01}$ |
| $B_{17}$ | $0.00 \pm .00$ | $0.03 \pm .00$ | $0.02 \pm .00$ | $0.31 \pm .02$ | $0.31 \pm .02$ | $0.32 \pm .02$ | $0.36 \pm .02$ | $0.38 \pm 0.04$ | $\mathbf{0.93 \pm 0.02}$ |
| $B_{18}$ | $0.01 \pm .00$ | $0.03 \pm .00$ | $0.03 \pm .00$ | $0.20 \pm .03$ | $0.22 \pm .01$ | $0.35 \pm .01$ | $0.38 \pm .01$ | $0.33 \pm 0.03$ | $\mathbf{0.86 \pm 0.03}$ |
| $B_{19}$ | $0.01 \pm .00$ | $0.02 \pm .00$ | $0.03 \pm .00$ | $0.26 \pm .02$ | $0.24 \pm .01$ | $0.48 \pm .01$ | $0.35 \pm .02$ | $0.39 \pm 0.01$ | $\mathbf{0.89 \pm 0.02}$ |
| $B_{20}$ | $0.02 \pm .00$ | $0.02 \pm .00$ | $0.04 \pm .00$ | $0.37 \pm .02$ | $0.34 \pm .03$ | $0.40 \pm .00$ | $0.34 \pm .02$ | $0.37 \pm 0.04$ | $\mathbf{0.83 \pm 0.02}$ |
| $B_{21}$ | $0.03 \pm .00$ | $0.03 \pm .00$ | $0.01 \pm .00$ | $0.11 \pm .01$ | $0.10 \pm .02$ | $0.22 \pm .02$ | $0.37 \pm .01$ | $0.14 \pm 0.02$ | $\mathbf{0.91 \pm 0.01}$ |
| | | | | | French | | | | |
| $B_1$ | $0.01 \pm 0.00$ | $0.03 \pm 0.00$ | $0.00 \pm .00$ | $0.28 \pm 0.02$ | $0.28 \pm 0.02$ | $0.33 \pm 0.01$ | $0.29 \pm 0.04$ | $0.30 \pm 0.04$ | $\mathbf{0.92 \pm 0.01}$ |
| $B_2$ | $0.04 \pm 0.00$ | $0.01 \pm 0.01$ | $0.01 \pm .00$ | $0.31 \pm 0.01$ | $0.30 \pm 0.03$ | $0.34 \pm 0.04$ | $0.33 \pm 0.03$ | $0.46 \pm 0.01$ | $\mathbf{0.94 \pm 0.04}$ |
| $B_3$ | $0.02 \pm 0.00$ | $0.01 \pm 0.01$ | $0.01 \pm .00$ | $0.35 \pm 0.01$ | $0.34 \pm 0.01$ | $0.37 \pm 0.01$ | $0.49 \pm 0.02$ | $0.51 \pm 0.02$ | $\mathbf{0.92 \pm 0.02}$ |
| $B_4$ | $0.03 \pm 0.00$ | $0.02 \pm 0.00$ | $0.01 \pm .00$ | $0.29 \pm 0.02$ | $0.43 \pm 0.02$ | $0.23 \pm 0.02$ | $0.29 \pm 0.03$ | $0.36 \pm 0.03$ | $\mathbf{0.90 \pm 0.01}$ |
| $B_5$ | $0.08 \pm 0.00$ | $0.02 \pm 0.01$ | $0.03 \pm .00$ | $0.37 \pm 0.02$ | $0.30 \pm 0.01$ | $0.34 \pm 0.01$ | $0.38 \pm 0.01$ | $0.40 \pm 0.03$ | $\mathbf{0.86 \pm 0.01}$ |
| $B_6$ | $0.03 \pm 0.00$ | $0.08 \pm 0.01$ | $0.01 \pm .00$ | $0.17 \pm 0.02$ | $0.20 \pm 0.03$ | $0.18 \pm 0.00$ | $0.40 \pm 0.03$ | $0.47 \pm 0.02$ | $\mathbf{0.93 \pm 0.02}$ |
| $B_7$ | $0.05 \pm 0.00$ | $0.10 \pm 0.00$ | $0.01 \pm .00$ | $0.29 \pm 0.02$ | $0.22 \pm 0.03$ | $0.23 \pm 0.01$ | $0.35 \pm 0.01$ | $0.47 \pm 0.01$ | $\mathbf{0.93 \pm 0.01}$ |
| $B_8$ | $0.05 \pm 0.00$ | $0.03 \pm 0.00$ | $0.02 \pm .00$ | $0.22 \pm 0.02$ | $0.22 \pm 0.01$ | $0.32 \pm 0.03$ | $0.38 \pm 0.03$ | $0.40 \pm 0.02$ | $\mathbf{0.89 \pm 0.01}$ |
| $B_9$ | $0.03 \pm 0.00$ | $0.01 \pm 0.01$ | $0.01 \pm .00$ | $0.22 \pm 0.02$ | $0.12 \pm 0.04$ | $0.18 \pm 0.01$ | $0.27 \pm 0.02$ | $0.37 \pm 0.01$ | $\mathbf{0.78 \pm 0.01}$ |
| $B_{10}$ | $0.07 \pm 0.00$ | $0.06 \pm 0.00$ | $0.02 \pm .00$ | $0.18 \pm 0.01$ | $0.19 \pm 0.01$ | $0.27 \pm 0.01$ | $0.40 \pm 0.04$ | $0.39 \pm 0.03$ | $\mathbf{0.87 \pm 0.02}$ |
| $B_{11}$ | $0.06 \pm 0.00$ | $0.02 \pm 0.00$ | $0.01 \pm .00$ | $0.16 \pm 0.02$ | $0.23 \pm 0.02$ | $0.18 \pm 0.03$ | $0.25 \pm 0.01$ | $0.35 \pm 0.02$ | $\mathbf{0.90 \pm 0.01}$ |
| $B_{12}$ | $0.08 \pm 0.00$ | $0.05 \pm 0.00$ | $0.02 \pm .00$ | $0.26 \pm 0.03$ | $0.23 \pm 0.04$ | $0.28 \pm 0.02$ | $0.57 \pm 0.02$ | $0.44 \pm 0.02$ | $\mathbf{0.90 \pm 0.02}$ |
| $B_{13}$ | $0.02 \pm 0.00$ | $0.01 \pm 0.00$ | $0.01 \pm .00$ | $0.17 \pm 0.01$ | $0.22 \pm 0.01$ | $0.20 \pm 0.03$ | $0.37 \pm 0.03$ | $0.64 \pm 0.03$ | $\mathbf{0.95 \pm 0.03}$ |
| $B_{14}$ | $0.02 \pm 0.00$ | $0.02 \pm 0.00$ | $0.03 \pm .00$ | $0.16 \pm 0.02$ | $0.33 \pm 0.02$ | $0.31 \pm 0.04$ | $0.54 \pm 0.02$ | $0.62 \pm 0.02$ | $\mathbf{0.94 \pm 0.02}$ |
| $B_{15}$ | $0.01 \pm 0.00$ | $0.04 \pm 0.01$ | $0.02 \pm .00$ | $0.24 \pm 0.04$ | $0.28 \pm 0.03$ | $0.36 \pm 0.02$ | $0.54 \pm 0.03$ | $0.48 \pm 0.02$ | $\mathbf{0.89 \pm 0.02}$ |

## Experiments and Results

**Data Sets:** We use two publicly available large-scale datasets Event2012 (English) (McMinn et al., 2013) and Event2018 (French) (Mazoyer et al., 2020), designed to evaluate social event detection. Event2012 comprises $68,841$ manually labelled tweets spanning 503 event classes over 29 days. Event2018 with $64,516$ labelled tweets across 257 event classes, covering 23 days, is used for cross-lingual experiments. The evolving social messages are split into blocks by date, according to the paper (Peng et al., 2023).

**Baselines and Evaluation Metrics:** We compare our model with BERT(Devlin et al., 2019), BiL-STM(Graves & Schmidhuber, 2005), EventX(Liu et al., 2020), KPGNN(Cao et al., 2021), FinEvent(Peng et al., 2023), CLKD(Ren et al., 2024) and Contextual (Liao et al., 2023). For complete details on the experimental settings, code and dataset, the reader can refer to the supplementary material. In alignment with prior studies (Cao et al., 2021), we utilise established metrics such as Adjusted Mutual Information (AMI), Adjusted Rand Index (ARI), and Normalised Mutual Information (NMI).

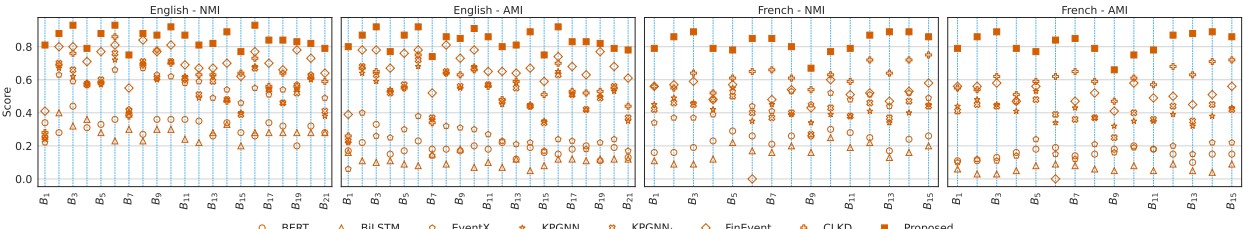

Figure 4: Dot plot comparing proposed model with some supervised models in terms of NMI and AMI.

## Results

Table 2 and Figure 4 show our results compared to supervised baselines in an online setting. We run our model 5 times with random seeds and report the average results. It is evident that our model consistently outperforms all supervised baselines across all metrics in both datasets. We see an average increase of 21.53% (NMI), 25.32% (AMI), and 146.25% (ARI) from the second-highest baseline for the English dataset. Similarly, an average increase of 26.57% (NMI), 29.15% (AMI), and 101.13% (ARI) is observed from the

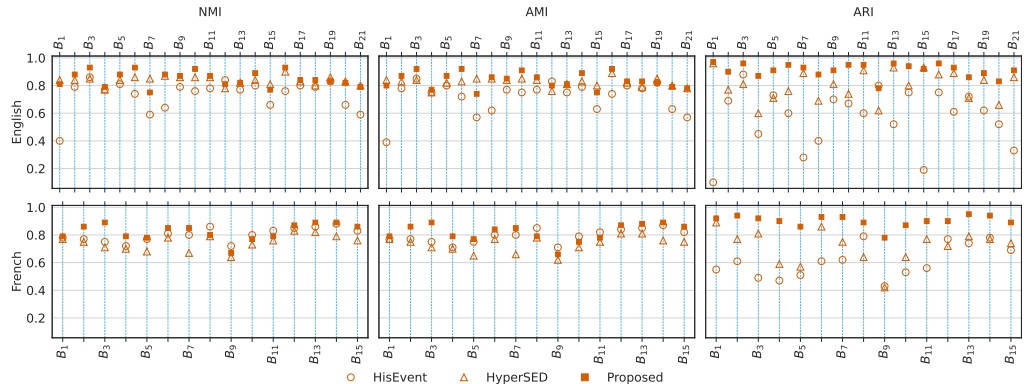

Figure 5: Dot plot comparing proposed supervised model with unsupervised models.

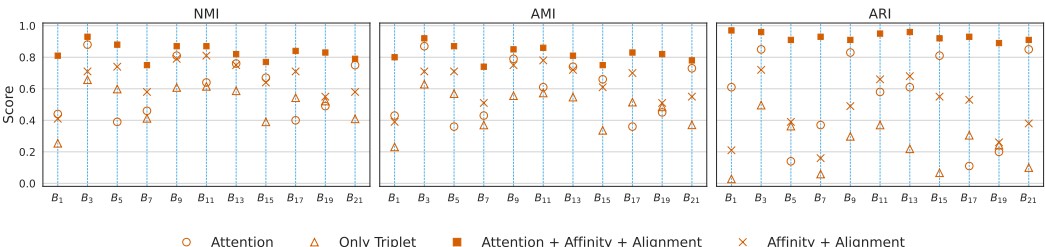

Figure 6: Dot plot showing the contribution of different components of our model for the English dataset.

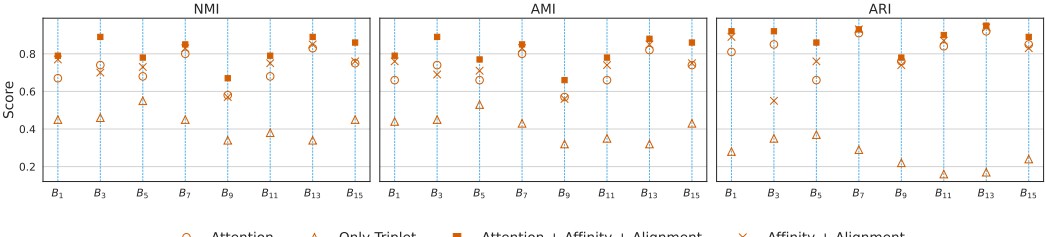

Figure 7: Dot plot showing the contribution of different components of our model for the French dataset.

second-highest baseline for the French dataset. The numerical results with standard deviation for NMI and AMI metrics are provided in the supplementary file. To validate our observations, we performed a statistical significance analysis using the Wilcoxon signed-rank test on the paired performance differences across all data blocks. The null hypothesis $(H_0)$ indicates that the median of the performance difference between our proposed method and the second-best supervised baseline method is zero. The alternative hypothesis $(H_1)$ is that this median is greater than zero, indicating our method's superiority. The test results, focusing on the combined NMI, AMI, and ARI metrics, confirm that our method's improved performance is statistically significant across both datasets. For the English dataset, we have $p < 0.00001$ and for the French dataset, we get $p \approx 0.00003$. In both cases, the null hypothesis is decisively rejected in favour of the alternative, indicating improved performance with our proposed method.

### Ablation Study

**Comparison with Unsupervised Methods:** We compare the proposed method with the unsupervised methods **HISEvent** Cao et al. (2024) and **HyperSED** Yu et al. (2025) and the results are shown in Figure 5. We can observe from the Figure 5 that for the English dataset, our method outperforms HISEvent and HyperSED across majority of blocks (ahead in $14, 15$ and $19$ blocks out of $21$ in terms of NMI, AMI and ARI

respectively), with average improvements of 3.88%, 4.09%, and 15.27% for NMI, AMI, and ARI, respectively. Similarly, for the French dataset, our method leads in most blocks, e.g., 11 in NMI and AMI, and all blocks in ARI, with average gains of 6.06% (NMI), 6.44% (AMI), and 26.47% (ARI). We perform a statistically significant test to validate the observations using the Wilcoxon signed-rank test. The null hypothesis ($H_0$) posits that the median of the performance difference between our method and a competitor is zero, while the alternative hypothesis ($H_1$) is that this median is greater than zero. The test results, focusing on the combined metric, confirm that our method's superiority is statistically significant across both datasets. For the English dataset, the proposed method shows a highly significant improvement over both HISEvent ($p < 0.00001$) and HyperSED ($p \approx 0.0003$). Similarly, for the French dataset, the method demonstrates a significant improvement over HISEvent ($p \approx 0.00006$) and HyperSED ($p \approx 0.00003$). Thus, in all cases, the null hypothesis is rejected in favour of the alternative, supporting our claim of superior performance. A slight decrease in our NMI and AMI scores is sometimes observed due to the operation of our model. It learns to group events by aligning them near each other in a shared label space. This approach can become a challenge when handling events that are separate but closely related, especially those that occur one after the other. Because of the connection between these consecutive events, they are mapped to adjacent regions in the label space. This closeness can cause the model to incorrectly merge the distinct events into a single cluster. This has a stronger negative impact on the NMI and AMI scores than on other metrics. However, this appears to be a minor trade-off, as our method's superior performance in most cases demonstrates its overall effectiveness.

**Impact of Individual Components:** To isolate the impact of our loss functions' components, we performed an ablation study comparing our final loss, $\mathcal{L}_{final}$, to the standard triplet loss ($\mathcal{L}_{trip}$) from Cao et al. (2021). The results for the English dataset are shown in Figure 6. Our attention module alone enhances the baseline triplet loss significantly, with average improvements of 21.80% (NMI), 27.14% (AMI), and 254.29% (ARI). In some blocks, however, its performance decreases, which we attribute to noise from heterogeneous interactions. Our proposed loss enhances the triplet loss by 27.20% in NMI, 30.63% in AMI, and 132.02% in ARI. By combining these elements, our complete loss function, $\mathcal{L}_{final}$, achieves the most substantial improvement, surpassing the baseline triplet loss by an average of 59.84% in NMI, 70.69% in AMI, and 470.95% in ARI. For the French dataset (Figure 7) the attention module enhances the baseline triplet loss by 73.44% (NMI), 79.30% (AMI), and 261.21% (ARI). Only our proposed loss enhances the triplet loss by 83.15% in NMI, 90.22% in AMI, and 266.57% in ARI. Combining all the elements ($\mathcal{L}_{final}$) we obtain an improvement of 100.08% (NMI), 108.55% (AMI), and 298.22% (ARI). Sensitivity analysis for hyperparameters are reported in Figures 11 and 12 in the supplementary.

**Generalizability of the Proposed Loss:** In order to assess the broader applicability of proposed Affinity and Alignment losses beyond the domain of social event detection, we evaluated their generalization capabilities against 25 different metric learning losses on four widely-used benchmark datasets: Cora (citation network), Amazon Computers and Amazon Photo (co-purchase networks), and Flickr (social network). The experiment learns discriminative representations using each loss in a standardized supervised training framework. We evaluated the quality of the learned embedding space by applying K-Means clustering and quantifying performance using NMI, AMI, and ARI. As detailed in Table 3, our method consistently demonstrates robust performance against a comprehensive suite of state-of-the-art metric learning baselines. Notably, it achieves significant gains on the Computers and Photo datasets, with average improvements of 6.42% and 7.32% respectively across all metrics, and secures a 10% increase in ARI for both. On the Flickr and Cora datasets, our approach remains highly competitive, matching or surpassing the majority of established baselines. These results empirically validate that the proposed Affinity and Alignment losses function effectively enforce feature consistency and separation on generalized tasks across diverse data domains.

**Failure Case Analysis:** We observed that for some of the cases, closely related events are merged into one. In order to understand the rationale, we analyzed one of the low NMI block, 'Block 15', of the English dataset. A visual comparison of the TSNE embeddings of our features with the ground truth and the predicted event classes are shown in Figure 8. We specifically focus on the predicted cluster (Cluster 11) (marked with the black circles in the right Figure) exhibiting the highest mixture of ground truth events. Our analysis reveals that there are noisy (spurious or missing) connections between these tweets based on the common attributes

Table 3: Generalization performance to standard metric learning benchmarks.

| Loss | Cora | | | Computers | | | Flickr | | | Photo | | |
|---|---|---|---|---|---|---|---|---|---|---|---|---|
| | NMI | AMI | ARI | NMI | AMI | ARI | NMI | AMI | ARI | NMI | AMI | ARI |
| TripletMargin (Balntas et al., 2016) | 0.55 | 0.54 | 0.51 | 0.62 | 0.62 | 0.62 | 0.01 | 0.01 | 0.00 | 0.78 | 0.77 | 0.70 |
| Contrastive (Musgrave et al., 2020) | 0.41 | 0.41 | 0.16 | 0.61 | 0.61 | 0.51 | 0.08 | 0.08 | 0.08 | 0.75 | 0.74 | 0.64 |
| MultiSimilarity (Wang et al., 2019) | 0.50 | 0.50 | 0.43 | 0.03 | 0.02 | 0.01 | 0.00 | 0.00 | 0.02 | 0.03 | 0.02 | 0.01 |
| Circle (Sun et al., 2020) | 0.56 | 0.55 | 0.53 | 0.51 | 0.51 | 0.27 | 0.01 | 0.01 | 0.00 | 0.66 | 0.65 | 0.49 |
| NTXent (Chen et al., 2020) | 0.57 | 0.56 | 0.53 | 0.57 | 0.57 | 0.42 | 0.01 | 0.01 | 0.00 | 0.74 | 0.74 | 0.62 |
| SupCon (Khosla et al., 2020) | 0.57 | 0.56 | 0.55 | 0.68 | 0.68 | 0.67 | 0.01 | 0.01 | 0.02 | 0.76 | 0.76 | 0.68 |
| LiftedStructure (Song et al., 2016) | 0.44 | 0.44 | 0.18 | 0.44 | 0.43 | 0.27 | 0.01 | 0.01 | 0.00 | 0.69 | 0.69 | 0.57 |
| NCA (Goldberger et al., 2004) | 0.50 | 0.49 | 0.30 | 0.56 | 0.56 | 0.46 | 0.01 | 0.01 | 0.00 | 0.79 | 0.79 | 0.74 |
| Margin (Wu et al., 2017) | 0.02 | 0.00 | 0.00 | 0.03 | 0.02 | 0.01 | 0.01 | 0.01 | 0.00 | 0.03 | 0.03 | 0.01 |
| SNR (Yuan et al., 2019) | 0.51 | 0.51 | 0.39 | 0.62 | 0.62 | 0.51 | 0.01 | 0.01 | 0.01 | 0.70 | 0.69 | 0.56 |
| FastAP (Farago, 2019) | **0.59** | **0.59** | **0.60** | 0.03 | 0.02 | 0.02 | 0.07 | 0.07 | 0.07 | 0.77 | 0.77 | 0.76 |
| GenLiftedStruct (Hermans et al., 2017) | 0.48 | 0.48 | 0.27 | 0.46 | 0.45 | 0.33 | 0.02 | 0.02 | 0.03 | 0.73 | 0.73 | 0.65 |
| Histogram (Ustinova & Lempitsky, 2016) | 0.53 | 0.52 | 0.48 | 0.38 | 0.37 | 0.39 | 0.08 | 0.08 | **0.09** | 0.77 | 0.77 | 0.69 |
| NPairs (Cogswell et al., 2016) | 0.50 | 0.49 | 0.37 | 0.03 | 0.02 | 0.01 | 0.01 | 0.01 | 0.00 | 0.03 | 0.02 | 0.01 |
| RankedList (Wang et al., 2022) | 0.53 | 0.53 | 0.46 | 0.56 | 0.55 | 0.39 | 0.01 | 0.01 | 0.00 | 0.74 | 0.74 | 0.64 |
| SmoothAP (Brown et al., 2020) | 0.05 | 0.04 | 0.02 | 0.12 | 0.12 | 0.06 | 0.04 | 0.04 | 0.06 | 0.30 | 0.30 | 0.17 |
| ArcFace (Deng et al., 2022) | 0.51 | 0.50 | 0.40 | 0.41 | 0.41 | 0.30 | 0.10 | 0.10 | 0.06 | 0.28 | 0.27 | 0.16 |
| CosFace (Wang et al., 2018) | 0.57 | 0.57 | 0.52 | 0.59 | 0.59 | 0.42 | 0.09 | 0.09 | 0.05 | 0.71 | 0.71 | 0.60 |
| LgMarginSoftmax (Liu et al., 2016) | 0.00 | 0.00 | 0.00 | 0.31 | 0.30 | 0.10 | 0.00 | 0.00 | 0.00 | 0.00 | 0.00 | 0.00 |
| ProxyAnchor (Kim et al., 2020) | 0.54 | 0.54 | 0.50 | 0.46 | 0.46 | 0.32 | 0.08 | 0.08 | 0.04 | 0.60 | 0.60 | 0.40 |
| ProxyNCA (Movshovitz-Attias et al., 2017) | 0.52 | 0.51 | 0.38 | 0.62 | 0.62 | 0.49 | 0.11 | 0.11 | 0.06 | 0.76 | 0.76 | 0.71 |
| SoftTriple (Qian et al., 2019) | 0.58 | 0.57 | 0.53 | 0.64 | 0.64 | 0.56 | 0.10 | 0.10 | 0.08 | 0.76 | 0.76 | 0.66 |
| P2SGrad (Zhang et al., 2019) | 0.50 | 0.50 | 0.29 | 0.54 | 0.53 | 0.31 | 0.07 | 0.07 | 0.05 | 0.73 | 0.73 | 0.63 |
| PNP (Li et al., 2022) | 0.58 | 0.57 | 0.57 | 0.62 | 0.62 | 0.52 | **0.11** | **0.11** | 0.07 | 0.74 | 0.74 | 0.63 |
| Contextual (Liao et al., 2023) | 0.53 | 0.53 | 0.45 | 0.33 | 0.33 | 0.21 | 0.01 | 0.01 | 0.00 | 0.51 | 0.50 | 0.38 |
| **Ours** | 0.56 | 0.56 | 0.58 | **0.71** | **0.71** | **0.74** | **0.11** | **0.11** | **0.09** | **0.83** | **0.83** | **0.85** |

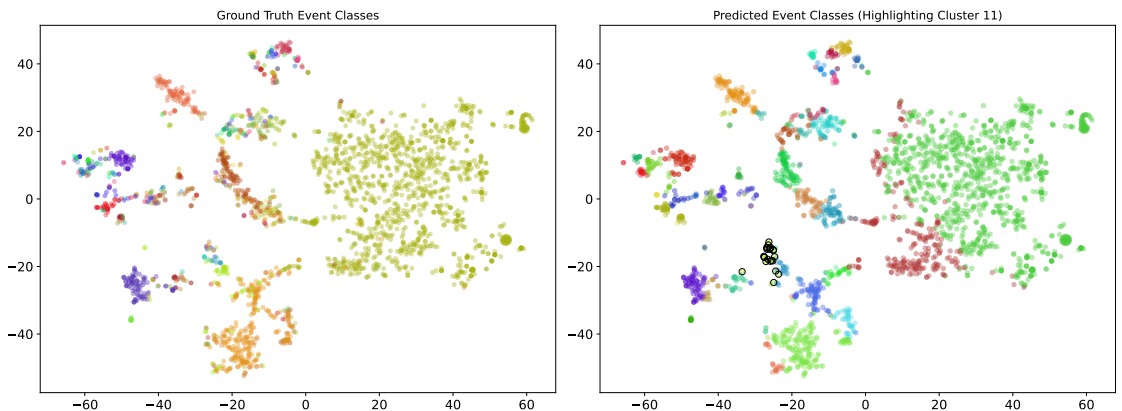

Figure 8: TSNE plots for Ground Truth Events and Predicted Events for the embedding of block $B_{15}$.

(entities, words and mentions) provided by the dataset. This can be seen in the Table 12 (supplementary) and the Figure 9b where we show the tweets of Cluster 11 and visualize the different relations between them respectively. Along with this, we also find that concatenating the timestamp encodings of a tweet with its text encodings increases the feature similarity between tweets of different events that arrive within a similar time frame. We highlight this in Figure 9a for the tweets in Cluster 11 which appear in the time window of a day.

## Conclusion

In this paper, we addressed the fundamental limitations of triplet-loss-based frameworks for social event detection. Through two novel regularising losses, namely, Affinity Loss and Alignment Loss, we resolve these challenges. Our approach ensures consistent feature separation by increasing intra-event affinity while aligning features to a common label space across minibatches. We have theoretically demonstrated that our joint loss optimisation is no less discriminative than CGML, while operating linearly over minibatches. The proposed attention module utilises heterogeneous attributes effectively.

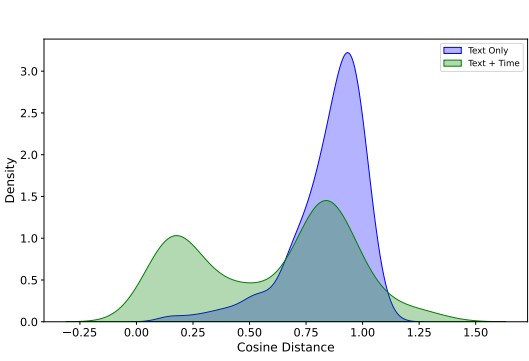

(a) Distribution of pairwise cosine distances

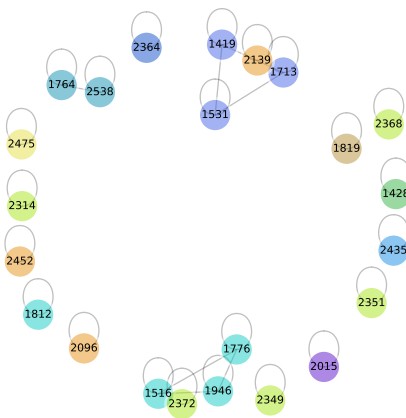

(b) Homogeneous layer graph for Cluster 11

Figure 9: Analysis of Cluster 11 in Block $B_{15}$.

The contributions of this work may be extended beyond event detection. For example, the proposed Affinity and Alignment losses can offer a robust solution for any deep metric learning task that relies on minibatch training and suffers from feature inconsistency.

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
