## Derivation of Distance Bounds for Inter-class Nodes

Let $x_i \in$ Class $c$ and $x_j \in$ Class $k$. Let $C_c, C_k$ be the class centers and $\pi = \|C_c - C_k\|_2$. Based on the definition of a Discriminative Graph, we assume the node-to-center distances are bounded by the radii: $\|x_i - C_c\|_2 \leq \frac{\alpha_c}{2}$ and $\|x_j - C_k\|_2 \leq \frac{\alpha_k}{2}$.

To find the maximum distance, we apply the standard Triangle Inequality:

$$\begin{aligned}
\|x_i - x_j\|_2 &= \|(x_i - C_c) + (C_c - C_k) + (C_k - x_j)\|_2 \\
&\leq \|x_i - C_c\|_2 + \|C_c - C_k\|_2 + \|C_k - x_j\|_2 \\
&\leq \frac{\alpha_c}{2} + \pi + \frac{\alpha_k}{2} = \frac{2\pi + \alpha_c + \alpha_k}{2}
\end{aligned}$$

To find the minimum distance, we apply the Reverse Triangle Inequality. For clarity, we use the absolute value to ensure the result is non-negative:

$$\begin{aligned}
\|x_i - x_j\|_2 &= \|(C_c - C_k) - (C_c - x_i) - (x_j - C_k)\|_2 \\
&\geq |\|C_c - C_k\|_2 - \|(C_c - x_i) + (x_j - C_k)\|_2| \\
&\geq |\|C_c - C_k\|_2 - (\|C_c - x_i\|_2 + \|x_j - C_k\|_2)|
\end{aligned}$$

Substituting the bounds $\pi$, $\alpha_c/2$, and $\alpha_k/2$:

$$\|x_i - x_j\|_2 \geq \left| \pi - \frac{\alpha_c + \alpha_k}{2} \right| = \frac{|2\pi - \alpha_c - \alpha_k|}{2}$$

Therefore, in a Discriminative Graph where $2\pi > \alpha_c + \alpha_k$, the distance is bounded by:

$$\|x_i - x_j\|_2 \in \left[ \frac{2\pi - \alpha_c - \alpha_k}{2}, \frac{2\pi + \alpha_c + \alpha_k}{2} \right] \tag{14}$$

## Proposition 1 and Proof (Necessary Condition)

**Proposition 1.** *Given Discriminative Graph $G$, randomly and independently sample $n$ data points from each class two times and thus obtain two data batches $X', X'' \in \mathbb{R}^{d \times nC}$ respectively, then construct the sub-graphs $G', G''$ along with adjacency matrices $S', S''$. We will have that $S' \approx S''$ is the necessary condition of a Discriminative Graph.*

**Proof of Necessary Condition**

**1. For the intra-class connected nodes** (taking for example the $c$-th class):
Since the exponential function is convex, from Jensen's Inequality, we have:

$$1 \geq E[s_{ij}] \geq \exp\left( -\frac{E[\|x_i - x_j\|_2^2]}{\sigma} \right) \geq \exp\left( -\frac{\alpha_c^2}{\sigma} \right)$$

Then, computing the expected squared difference:

$$\begin{aligned}
E[(s_{ij}' - s_{ij}'')^2] &= E[((s_{ij}' - E[s_{ij}]) - (s_{ij}'' - E[s_{ij}]))^2] \\
&= E[(s_{ij}' - E[s_{ij}])^2 + (s_{ij}'' - E[s_{ij}])^2 - 2(s_{ij}' - E[s_{ij}])(s_{ij}'' - E[s_{ij}])]
\end{aligned}$$

Since the data points are i.i.d., $E[s_{ij}] = E[s_{ij}'] = E[s_{ij}'']$ and $E[s_{ij} - E[s_{ij}]] = 0$, thus:

$$= 2E[(s_{ij} - E[s_{ij}])^2] = 2Var(s_{ij}) = 2(E[s_{ij}^2] - E^2[s_{ij}]) \leq 2(1 - \exp(-\frac{2\alpha_c^2}{\sigma}))$$

The upper bound $2(1 - \exp(-\frac{2\alpha_c^2}{\sigma}))$ is proportional to $\alpha_c$, indicating that the differences between $s'_{ij}$ and $s''_{ij}$ are consistent with intra-class compactness. When $\alpha_c$ is small the expected squared difference is bounded by a very small value, meaning $s'_{ij} \approx s''_{ij}$.

**2. For the inter-class connected nodes** (where $x_i, x_j$ are sampled from different classes $c$ and $k$):
Let $\pi$ be the distance between the $c$-th and $k$-th class centers, where $\alpha_c < \pi$ and $\alpha_k < \pi$. The similarity $s_{ij}$ lies within the interval (from Equation 14):

$$[a, b] = \left[ \exp\left( -\frac{(\alpha_c + \alpha_k + 2\pi)^2}{4\sigma} \right), \exp\left( -\frac{(2\pi - \alpha_c - \alpha_k)^2}{4\sigma} \right) \right]$$

Using Hoeffding's Inequality, we have:

$$Pr\{|s_{ij} - E[s_{ij}]| \geq t\} \leq 2\exp\left( -\frac{2t^2}{(b-a)^2} \right)$$

To find the bound $t$ that holds with a confidence level of $1 - \delta/2$, we set the probability of failure to $\delta/2$:

$$\delta/2 = 2\exp\left( -\frac{2t^2}{(b-a)^2} \right)$$

Solving for $t$:

$$\frac{\delta}{4} = \exp\left( -\frac{2t^2}{(b-a)^2} \right)$$
$$\ln\left( \frac{4}{\delta} \right) = \frac{2t^2}{(b-a)^2}$$
$$t^2 = \frac{(b-a)^2 \ln(4/\delta)}{2}$$
$$t = (b-a)\sqrt{\frac{\ln(4/\delta)}{2}}$$

Thus, with probability at least $1 - \delta/2$, we have $|s_{ij} - E(s_{ij})| \leq (b-a)\sqrt{\frac{\ln(4/\delta)}{2}}$

$$|s_{ij} - E[s_{ij}]| \leq (b-a)\sqrt{\frac{\log(4/\delta)}{2}}$$

Since $s'_{ij}, s''_{ij}$ are i.i.d., we have:

$$|s'_{ij} - s''_{ij}| \leq |s'_{ij} - E[s'_{ij}]| + |s''_{ij} - E[s''_{ij}]| \leq 2(b-a)\sqrt{\frac{\log(4/\delta)}{2}}$$

The upper bound of the absolute difference is proportional to $\alpha_c, \alpha_k$ (intra-class compactness) and inversely proportional to $\pi$ (inter-class separability). In both intra-class and inter-class cases the upper bounds are proportional to intra-class distance. For a Discriminative Graph, these upper bounds are less, leading to $s'_{ij} \approx s''_{ij}$, or $S' \approx S''$.

## Supporting proof of Lemmas

**Lemma 6.** *For any two matrices $A, B \in \mathbb{R}^{m \times n}$, the following inequality holds:*

$$\|A - B\|_F^2 \leq 2\|A\|_F^2 + 2\|B\|_F^2$$

*Proof.* From the triangle inequality, for the Frobenius norm:

$$\|X + Y\|_F \leq \|X\|_F + \|Y\|_F$$

By setting $X = A$ and $Y = -B$, we can write:

$$\|A - B\|_F = \|A + (-B)\|_F \leq \|A\|_F + \|-B\|_F$$

Given that $\|-B\|_F = \|B\|_F$, the inequality simplifies to:

$$\|A - B\|_F \leq \|A\|_F + \|B\|_F$$

Both sides being non-negative we have:

$$\|A - B\|_F^2 \leq (\|A\|_F + \|B\|_F)^2 = \|A\|_F^2 + 2\|A\|_F\|B\|_F + \|B\|_F^2$$

From inequality $2xy \leq x^2 + y^2$ we have:

$$2\|A\|_F\|B\|_F \leq \|A\|_F^2 + \|B\|_F^2$$

Therefore we get:

$$\|A\|_F^2 + 2\|A\|_F\|B\|_F + \|B\|_F^2 \leq \|A\|_F^2 + (\|A\|_F^2 + \|B\|_F^2) + \|B\|_F^2$$

Simplifying the expression on the right gives:

$$\|A\|_F^2 + 2\|A\|_F\|B\|_F + \|B\|_F^2 \leq 2\|A\|_F^2 + 2\|B\|_F^2$$

Using above inequality we get:

$$\|A - B\|_F^2 \leq \|A\|_F^2 + 2\|A\|_F\|B\|_F + \|B\|_F^2 \leq 2\|A\|_F^2 + 2\|B\|_F^2$$

This proves:

$$\|A - B\|_F^2 \leq 2\|A\|_F^2 + 2\|B\|_F^2$$

$\square$

**Lemma 7.** *Let $A$, $B$, and $C$ be matrices of the same dimensions (e.g., in $\mathbb{R}^{m \times n}$). The following inequality holds for the Frobenius norm $\|\cdot\|_F$:*

$$\|A - B\|_F \leq \|A - C\|_F + \|B - C\|_F$$

*Proof.* For any matrix $C$ we have:
$$A - B = (A - C) + (C - B)$$

Now, taking Frobenius norm:
$$\|A - B\|_F = \|(A - C) + (C - B)\|_F$$

By triangle inequality, we get:

$$\|(A - C) + (C - B)\|_F \leq \|A - C\|_F + \|C - B\|_F$$

$$\|(A - C) + (C - B)\|_F \leq \|A - C\|_F + \|B - C\|_F$$

$$(\because \|-M\| = \|M\|)$$

$$\therefore \|A - B\|_F \leq \|A - C\|_F + \|B - C\|_F$$

This completes the proof. $\square$

**Proof of Lemma 4:**

*Proof.* We start with the identity:

$$\|S'' - S'\|_F^2 = \|(S'' - L) - (S' - L)\|_F^2.$$

Using the inequality $\|A - B\|_F^2 \leq 2\|A\|_F^2 + 2\|B\|_F^2$, we get:

$$\|S'' - S'\|_F^2 \leq 2\|S'' - L\|_F^2 + 2\|S' - L\|_F^2.$$

Now take the limit of both sides:

$$\lim \|S'' - S'\|_F^2 \leq 2\lim \|S'' - L\|_F^2 + 2\lim \|S' - L\|_F^2 \to 0.$$

Since squared norms are non-negative, this implies $\lim \|S'' - S'\|_F^2 \to 0$.

$\square$

## Practical Running Time

The initial feature generation and incremental block generation take (280.16, 100.85) seconds and (130.58, 47.00) for the English and French datasets, respectively. The time of training for a minibatch when we consider all blocks are shown in Figure 10a and 10b for the English and the French dataset, respectively. We run all our experiments on a system with Ubuntu 20.04.5 LTS with an AMD EPYC 7282 16-Core processor and NVIDIA A30 GPU with 256 GB ram.

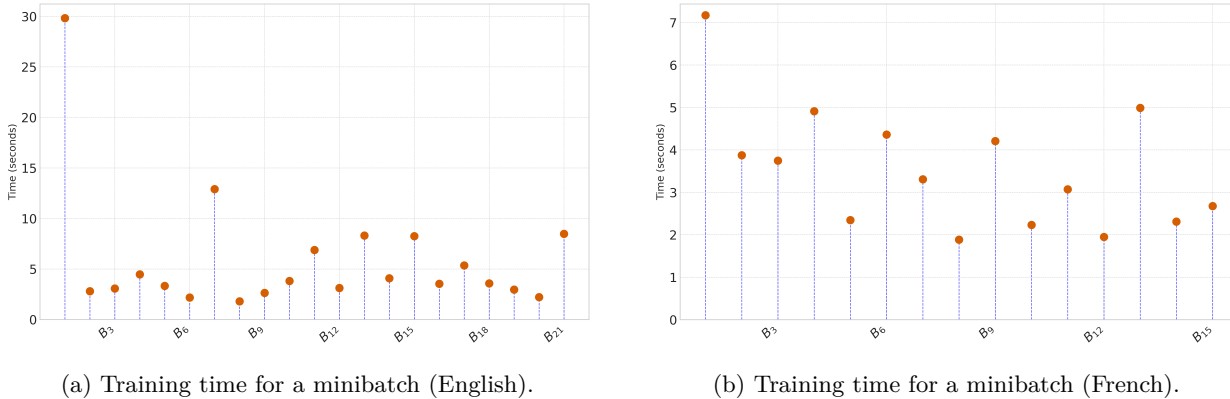

(a) Training time for a minibatch (English).      (b) Training time for a minibatch (French).

Figure 10: Comparison of training time for a minibatch across languages.

## Experiment Settings and Results

We have divided the tweets into blocks using the code from offical KPGNN repository for all the baselines.

**BERT:** The results for BERT are obtained after clustering the BERT encodings ('base-multilingual-cased') of both datasets with the DBSCAN clustering method to keep it unsupervised. We use the respective language encoders for each data set (English for Event2012 and French for Event2018).

**BiLSTM:** The BiLSTM model is trained using Triplet Loss with a learning rate of 0.001, batch size of 1000, dropout of 0.8, output dimension of 64, one LSTM bidirectional layer, 20 epochs and K-Means with the actual number of clusters. We use K-Means here as it gives better results.

**EventX:** We train this with a minimum number of co-occurrences threshold set to 2, conditional probability threshold for occurrence of words set to 0.15 and the minimum number of node threshold to stop graph splitting set to 3 as suggested in Cao et al. (2021) and Liu et al. (2020).

**KPGNN and Ours:** For training KPGNN, we utilized the incremental "Latest Message Strategy" (code setting `remove_obsolete=2`), where inference relies solely on the current block's data, and model maintenance occurs every `window_size=3` blocks using the most recent message block for training. During training phases, 20% of indices are reserved for validation (`validation_percent=0.2`) to monitor early stopping with a patience of 5 epochs. The model architecture features a Graph Attention Network (GAT) backbone with residual connections and 4 attention heads, processing a homogeneous graph. Training uses a batch size of 1000 and a learning rate of 0.001 for 15 epochs, optimizing a Triplet (`loss_setting=1` in our code) loss (margin 3.0). The node representations use a hidden dimension of 8 and an output dimension of 32, with 800 neighbors sampled per node. All these parameters are adapted from the official implementation of the KPGNN paper (Cao et al., 2021). For clustering, K-Means is used with the actual number of clusters, as this produces the best results and is the default choice for the default KPGNN implementation. The $KPGNN_t$ also uses the global-local pair loss mentioned in the paper (Cao et al., 2021). In our case keeping the rest of the parameters same as mentioned previously, we use the composite loss function (`loss_setting=3` in our code) that sums Triplet (margin 3.0), Affinity, and Alignment losses with our custom attention function. The link to the code repository is: `https://github.com/Shraban123/DeepAlign-IED/`.

**FinEvent:** In our experiments, we get the best results when we use a window size of 3, batch size of 1000, learning rate of 0.001, GAT model with 4 attention heads and residual connections and output dimension of 64. The number of epochs is 15. The step size of RL-0 for state1 and state3 is 0.02. The initial value of epsilon for state2 is set as 0.001 with a step size of 0.02. All the parameters are taken from the author's implementation with the paper Peng et al. (2023) combined with the latest message strategy as mentioned earlier. In the case of the Event2018 dataset, we use the $FinEvent_g$ setting reported in the paper.

**CLKD:** In our experiments, we get the best results when we train for 15 epochs using a window size of 3 (here also we use the latest message strategy), batch size of 1000, learning rate of 0.001, latest message strategy, output dimension of 32, GAT model with 4 attention heads and residual connections. For the Event2012 dataset, we use mode 1 as the teacher and student are in the same language. For Event2018, we report the results on the mode 2 with linear cross-lingual knowledge distillation with English as the teacher model and French as the student model.

**HISEvent and HyperSED:** We use the minimum group $n = 10$ for smaller initial clusters that have the possibility to merge into bigger clusters. The choice of n does not affect the performance of HISEvent significantly, as shown in the paper, but taking a lower value of n avoids the deadlock situation as suggested by Yu et al. (2024). We use the results of HyperSED reported in the paper.

**Additional results:** We present the tabular results for unsupervised methods in Tables 6 - 11. HISEvent shows no deviation with the specified setting. The tabular results for supervised methods in terms of AMI and NMI are in Tables 4 and 5, respectively. Also we report the sensitivity to hyperparameters $\sigma$ and $m_1, m_2$ in Figures 11 and 12 respectively. We can see from the results that a values of $\sigma$ smaller than 1.0 produces worse results across the metrics for both the datasets. For the parameters $m_1$ and $m_2$, one can see that the model performs poorly when the inter class margin ($m_2$) is smaller or equivalent to that of the intra class margin ($m_1$) which is expected and setting $m_1 = 1$ and $m_2 = 0$ gives the best results.

## Ethical Statement and Data Privacy

To ensure strict adherence to the X (formerly Twitter) Developer Agreement and Policy, this study does not redistribute raw tweet content, associated metadata, or personally identifiable information. Instead, our methodology utilizes derived, non-reversible representations of the data. Specifically, the datasets are processed into adjacency matrices stored in .npz format—and high-dimensional text embeddings. These formats encapsulate the structural relationships and semantic features necessary for social event detection without exposing the original proprietary text with user details. This approach ensures that the work remains fully reproducible for the scientific community while maintaining the privacy and security of the underlying data. It must be noted that we only list very few examples of this data (decoupled from user information) from existing public repositories for ease of understanding and qualitative study without ant redistribution.

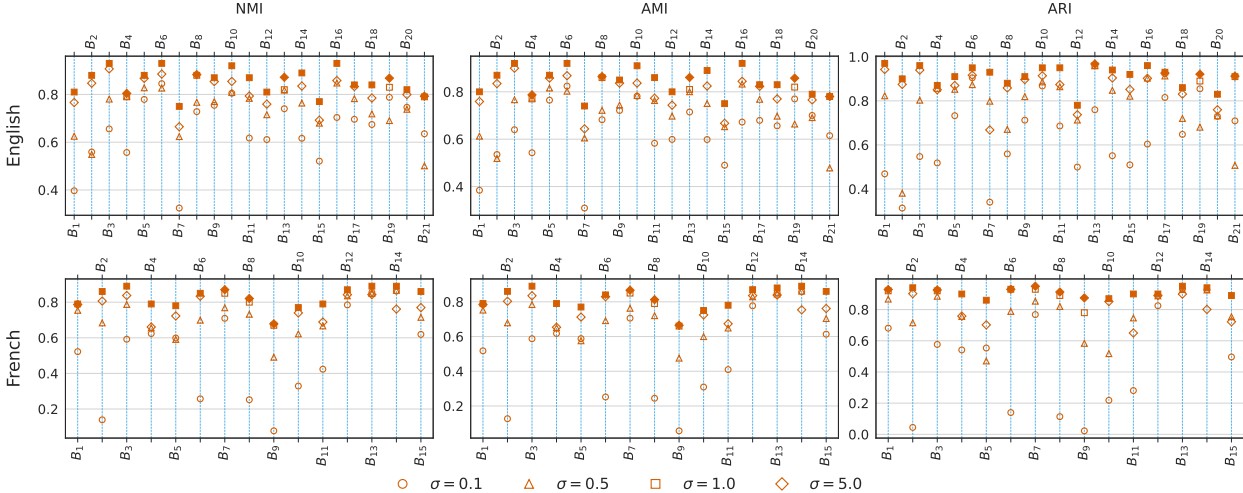

Figure 11: Ablation study on $\sigma$ for both the English and French datasets. Symbol for the the highest values are filled.

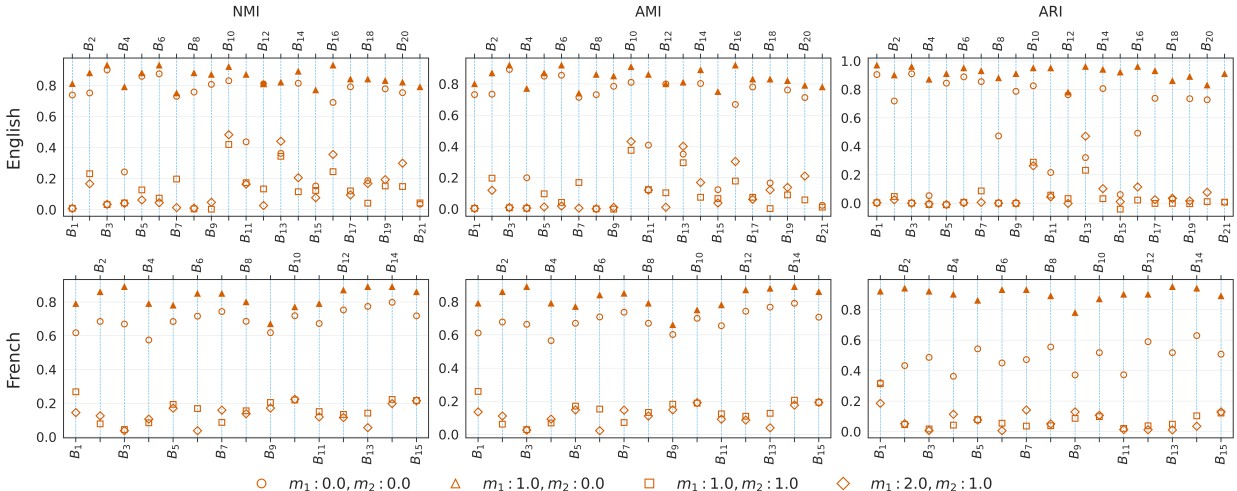

Figure 12: Ablation study on $m_1$, $m_2$ for both the English and French datasets. Symbol for the the highest values are filled.

These datasets are employed exclusively for non-commercial, academic research purposes, with a specific focus on the development and evaluation of social event detection algorithms. By using these processed feature vectors and graph structures rather than raw content, we contribute to the advancement of the area without violating the proprietary rights of the data source. Our commitment to ethical data practices is central to this work, and we provide these derived representations to facilitate peer review and further study while remaining in full compliance with all relevant platform policies.

Table 4: Event detection performance for supervised methods in terms of AMI across different blocks.

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

Table 5: Event detection performance for supervised methods in terms of NMI across different blocks.

| Block | BERT | BiLSTM | EventX | KPGNN | KPGNN$_t$ | FinEvent$_k$ | CLKD | Contextual | **Proposed** |
|---|---|---|---|---|---|---|---|---|---|
| | | | | | English | | | | |
| $B_1$ | $0.34 \pm .00$ | $0.27 \pm .00$ | $0.22 \pm .00$ | $0.24 \pm .01$ | $0.25 \pm .00$ | $0.40 \pm .00$ | $0.28 \pm .04$ | $0.31 \pm 0.01$ | **$0.81 \pm 0.02$** |
| $B_2$ | $0.28 \pm .00$ | $0.40 \pm .00$ | $0.63 \pm .00$ | $0.67 \pm .02$ | $0.70 \pm .01$ | $0.80 \pm .00$ | $0.69 \pm .01$ | $0.71 \pm 0.03$ | **$0.88 \pm 0.01$** |
| $B_3$ | $0.44 \pm .00$ | $0.32 \pm .00$ | $0.59 \pm .00$ | $0.62 \pm .01$ | $0.66 \pm .01$ | $0.80 \pm .00$ | $0.76 \pm .00$ | $0.77 \pm 0.04$ | **$0.93 \pm 0.04$** |
| $B_4$ | $0.31 \pm .00$ | $0.36 \pm .00$ | $0.58 \pm .00$ | $0.58 \pm .01$ | $0.57 \pm .01$ | $0.68 \pm .01$ | $0.57 \pm .02$ | $0.64 \pm 0.04$ | **$0.79 \pm 0.02$** |
| $B_5$ | $0.33 \pm .00$ | $0.28 \pm .00$ | $0.58 \pm .00$ | $0.57 \pm .01$ | $0.60 \pm .00$ | $0.73 \pm .01$ | $0.58 \pm .01$ | $0.66 \pm 0.01$ | **$0.88 \pm 0.01$** |
| $B_6$ | $0.36 \pm .00$ | $0.23 \pm .00$ | $0.66 \pm .00$ | $0.72 \pm .00$ | $0.76 \pm .00$ | $0.81 \pm .00$ | $0.86 \pm .00$ | $0.77 \pm 0.04$ | **$0.93 \pm 0.01$** |
| $B_7$ | $0.39 \pm .00$ | $0.30 \pm .00$ | $0.42 \pm .00$ | $0.40 \pm .00$ | $0.41 \pm .02$ | $0.52 \pm .02$ | $0.38 \pm .02$ | $0.47 \pm 0.03$ | **$0.75 \pm 0.01$** |
| $B_8$ | $0.27 \pm .00$ | $0.23 \pm .00$ | $0.67 \pm .00$ | $0.70 \pm .01$ | $0.71 \pm .01$ | $0.82 \pm .01$ | $0.69 \pm .00$ | $0.74 \pm 0.03$ | **$0.88 \pm 0.02$** |
| $B_9$ | $0.36 \pm .00$ | $0.30 \pm .00$ | $0.63 \pm .00$ | $0.60 \pm .02$ | $0.61 \pm .01$ | $0.73 \pm .00$ | $0.78 \pm .01$ | $0.67 \pm 0.02$ | **$0.87 \pm 0.02$** |
| $B_{10}$ | $0.36 \pm .00$ | $0.30 \pm .00$ | $0.62 \pm .00$ | $0.71 \pm .00$ | $0.71 \pm .00$ | $0.81 \pm .00$ | $0.70 \pm .01$ | $0.72 \pm 0.02$ | **$0.92 \pm 0.03$** |
| $B_{11}$ | $0.36 \pm .00$ | $0.24 \pm .00$ | $0.58 \pm .00$ | $0.62 \pm .01$ | $0.61 \pm .01$ | $0.69 \pm .02$ | $0.60 \pm .02$ | $0.61 \pm 0.02$ | **$0.87 \pm 0.01$** |
| $B_{12}$ | $0.35 \pm .00$ | $0.22 \pm .00$ | $0.59 \pm .00$ | $0.49 \pm .01$ | $0.51 \pm .01$ | $0.67 \pm .02$ | $0.63 \pm .00$ | $0.50 \pm 0.01$ | **$0.81 \pm 0.02$** |
| $B_{13}$ | $0.26 \pm .00$ | $0.28 \pm .00$ | $0.49 \pm .00$ | $0.62 \pm .00$ | $0.59 \pm .00$ | $0.67 \pm .00$ | $0.63 \pm .00$ | $0.63 \pm 0.03$ | **$0.82 \pm 0.02$** |
| $B_{14}$ | $0.34 \pm .00$ | $0.33 \pm .00$ | $0.54 \pm .00$ | $0.48 \pm .01$ | $0.48 \pm .00$ | $0.70 \pm .01$ | $0.47 \pm .02$ | $0.60 \pm 0.02$ | **$0.89 \pm 0.02$** |
| $B_{15}$ | $0.28 \pm .00$ | $0.20 \pm .00$ | $0.46 \pm .00$ | $0.40 \pm .01$ | $0.39 \pm .01$ | $0.59 \pm .02$ | $0.64 \pm .00$ | $0.47 \pm 0.02$ | **$0.77 \pm 0.03$** |
| $B_{16}$ | $0.26 \pm .00$ | $0.28 \pm .00$ | $0.55 \pm .00$ | $0.68 \pm .01$ | $0.67 \pm .01$ | $0.75 \pm .01$ | $0.73 \pm .03$ | $0.77 \pm 0.03$ | **$0.93 \pm 0.02$** |
| $B_{17}$ | $0.34 \pm .00$ | $0.28 \pm .00$ | $0.51 \pm .00$ | $0.54 \pm .01$ | $0.54 \pm .01$ | $0.70 \pm .00$ | $0.56 \pm .00$ | $0.69 \pm 0.04$ | **$0.84 \pm 0.02$** |
| $B_{18}$ | $0.32 \pm .00$ | $0.28 \pm .00$ | $0.54 \pm .00$ | $0.46 \pm .00$ | $0.46 \pm .00$ | $0.64 \pm .02$ | $0.64 \pm .02$ | $0.58 \pm 0.03$ | **$0.84 \pm 0.01$** |
| $B_{19}$ | $0.20 \pm .00$ | $0.27 \pm .00$ | $0.57 \pm .00$ | $0.54 \pm .00$ | $0.52 \pm .01$ | $0.74 \pm .02$ | $0.56 \pm .02$ | $0.60 \pm 0.01$ | **$0.83 \pm 0.01$** |
| $B_{20}$ | $0.32 \pm .00$ | $0.28 \pm .00$ | $0.63 \pm .00$ | $0.60 \pm .01$ | $0.62 \pm .01$ | $0.71 \pm .01$ | $0.61 \pm .01$ | $0.63 \pm 0.04$ | **$0.82 \pm 0.01$** |
| $B_{21}$ | $0.28 \pm .00$ | $0.27 \pm .00$ | $0.49 \pm .00$ | $0.38 \pm .00$ | $0.41 \pm .01$ | $0.61 \pm .01$ | $0.59 \pm .02$ | $0.48 \pm 0.02$ | **$0.79 \pm 0.01$** |
| | | | | | French | | | | |
| $B_1$ | $0.16 \pm 0.00$ | $0.11 \pm 0.01$ | $0.34 \pm 0.00$ | $0.45 \pm 0.02$ | $0.42 \pm 0.02$ | $0.56 \pm 0.01$ | $0.56 \pm 0.04$ | $0.59 \pm 0.04$ | **$0.79 \pm 0.02$** |
| $B_2$ | $0.16 \pm 0.00$ | $0.09 \pm 0.01$ | $0.37 \pm 0.00$ | $0.49 \pm 0.01$ | $0.46 \pm 0.03$ | $0.57 \pm 0.04$ | $0.55 \pm 0.03$ | $0.65 \pm 0.01$ | **$0.86 \pm 0.02$** |
| $B_3$ | $0.19 \pm 0.00$ | $0.09 \pm 0.01$ | $0.37 \pm 0.01$ | $0.46 \pm 0.01$ | $0.45 \pm 0.01$ | $0.59 \pm 0.01$ | $0.64 \pm 0.02$ | $0.68 \pm 0.02$ | **$0.89 \pm 0.02$** |
| $B_4$ | $0.23 \pm 0.00$ | $0.12 \pm 0.01$ | $0.39 \pm 0.00$ | $0.42 \pm 0.02$ | $0.48 \pm 0.02$ | $0.48 \pm 0.02$ | $0.52 \pm 0.03$ | $0.57 \pm 0.03$ | **$0.79 \pm 0.01$** |
| $B_5$ | $0.29 \pm 0.00$ | $0.22 \pm 0.00$ | $0.53 \pm 0.01$ | $0.55 \pm 0.02$ | $0.50 \pm 0.01$ | $0.57 \pm 0.01$ | $0.61 \pm 0.01$ | $0.60 \pm 0.03$ | **$0.78 \pm 0.04$** |
| $B_6$ | $0.26 \pm 0.00$ | $0.17 \pm 0.01$ | $0.44 \pm 0.00$ | $0.35 \pm 0.02$ | $0.40 \pm 0.03$ | $0.51 \pm 0.00$ | $0.65 \pm 0.03$ | $0.62 \pm 0.02$ | **$0.85 \pm 0.02$** |
| $B_7$ | $0.21 \pm 0.00$ | $0.16 \pm 0.01$ | $0.41 \pm 0.01$ | $0.45 \pm 0.02$ | $0.37 \pm 0.03$ | $0.48 \pm 0.01$ | $0.66 \pm 0.01$ | $0.68 \pm 0.01$ | **$0.85 \pm 0.02$** |
| $B_8$ | $0.26 \pm 0.00$ | $0.20 \pm 0.00$ | $0.53 \pm 0.01$ | $0.39 \pm 0.02$ | $0.40 \pm 0.01$ | $0.54 \pm 0.03$ | $0.61 \pm 0.03$ | $0.58 \pm 0.02$ | **$0.80 \pm 0.01$** |
| $B_9$ | $0.26 \pm 0.00$ | $0.16 \pm 0.00$ | $0.45 \pm 0.01$ | $0.34 \pm 0.02$ | $0.27 \pm 0.04$ | $0.43 \pm 0.01$ | $0.54 \pm 0.02$ | $0.54 \pm 0.01$ | **$0.67 \pm 0.01$** |
| $B_{10}$ | $0.30 \pm 0.00$ | $0.25 \pm 0.01$ | $0.52 \pm 0.00$ | $0.39 \pm 0.01$ | $0.43 \pm 0.01$ | $0.60 \pm 0.01$ | $0.63 \pm 0.04$ | $0.61 \pm 0.03$ | **$0.77 \pm 0.02$** |
| $B_{11}$ | $0.28 \pm 0.00$ | $0.19 \pm 0.00$ | $0.48 \pm 0.01$ | $0.38 \pm 0.02$ | $0.38 \pm 0.02$ | $0.51 \pm 0.03$ | $0.59 \pm 0.01$ | $0.61 \pm 0.02$ | **$0.79 \pm 0.03$** |
| $B_{12}$ | $0.25 \pm 0.00$ | $0.22 \pm 0.00$ | $0.51 \pm 0.01$ | $0.41 \pm 0.03$ | $0.46 \pm 0.04$ | $0.52 \pm 0.02$ | $0.72 \pm 0.02$ | $0.62 \pm 0.02$ | **$0.87 \pm 0.01$** |
| $B_{13}$ | $0.17 \pm 0.00$ | $0.13 \pm 0.00$ | $0.44 \pm 0.00$ | $0.34 \pm 0.01$ | $0.37 \pm 0.01$ | $0.47 \pm 0.03$ | $0.64 \pm 0.03$ | $0.76 \pm 0.03$ | **$0.89 \pm 0.01$** |
| $B_{14}$ | $0.24 \pm 0.00$ | $0.16 \pm 0.00$ | $0.52 \pm 0.00$ | $0.40 \pm 0.02$ | $0.47 \pm 0.02$ | $0.53 \pm 0.04$ | $0.72 \pm 0.02$ | $0.73 \pm 0.02$ | **$0.89 \pm 0.01$** |
| $B_{15}$ | $0.26 \pm 0.00$ | $0.20 \pm 0.01$ | $0.49 \pm 0.00$ | $0.45 \pm 0.04$ | $0.44 \pm 0.03$ | $0.58 \pm 0.02$ | $0.75 \pm 0.03$ | $0.63 \pm 0.02$ | **$0.86 \pm 0.01$** |

Table 6: ARI: Proposed vs. Unsupervised on English data.

| | $B_1$ | $B_2$ | $B_3$ | $B_4$ | $B_5$ | $B_6$ | $B_7$ |
|---|---|---|---|---|---|---|---|
| HISEvent | 0.10 | 0.69 | 0.88 | 0.45 | 0.73 | 0.6 | 0.28 |
| HyperSED | 0.96 | 0.77 | 0.81 | 0.60 | 0.71 | 0.76 | 0.89 |
| Proposed | **0.97** | **0.89** | **0.96** | **0.87** | **0.91** | **0.95** | **0.93** |
| | $B_8$ | $B_9$ | $B_{10}$ | $B_{11}$ | $B_{12}$ | $B_{13}$ | $B_{14}$ |
| HISEvent | 0.4 | 0.7 | 0.67 | 0.6 | 0.8 | 0.52 | 0.75 |
| HyperSED | 0.69 | 0.81 | 0.74 | 0.91 | 0.62 | 0.93 | 0.80 |
| Proposed | **0.88** | **0.91** | **0.95** | **0.95** | **0.78** | **0.96** | **0.94** |
| | $B_{15}$ | $B_{16}$ | $B_{17}$ | $B_{18}$ | $B_{19}$ | $B_{20}$ | $B_{21}$ |
| HISEvent | 0.19 | 0.75 | 0.61 | 0.72 | 0.62 | 0.52 | 0.33 |
| HyperSED | **0.93** | 0.88 | 0.89 | 0.71 | 0.84 | 0.66 | 0.86 |
| Proposed | 0.92 | **0.96** | **0.93** | **0.79** | **0.89** | **0.83** | **0.91** |

Table 7: AMI: Proposed vs. Unsupervised on English data.

| | $B_1$ | $B_2$ | $B_3$ | $B_4$ | $B_5$ | $B_6$ | $B_7$ |
|---|---|---|---|---|---|---|---|
| HISEvent | 0.39 | 0.78 | 0.85 | 0.75 | 0.80 | 0.72 | 0.57 |
| HyperSED | 0.84 | 0.83 | 0.84 | 0.75 | 0.82 | 0.83 | **0.85** |
| Proposed | **0.80** | **0.85** | **0.91** | **0.77** | **0.87** | **0.92** | 0.74 |
| | $B_8$ | $B_9$ | $B_{10}$ | $B_{11}$ | $B_{12}$ | $B_{13}$ | $B_{14}$ |
| HISEvent | 0.62 | 0.77 | 0.75 | 0.77 | **0.83** | 0.75 | 0.79 |
| HyperSED | 0.85 | 0.84 | 0.85 | 0.84 | 0.76 | 0.81 | 0.83 |
| Proposed | **0.86** | **0.85** | **0.89** | **0.86** | 0.74 | 0.81 | **0.89** |
| | $B_{15}$ | $B_{16}$ | $B_{17}$ | $B_{18}$ | $B_{19}$ | $B_{20}$ | $B_{21}$ |
| HISEvent | 0.63 | 0.74 | 0.8 | 0.78 | 0.82 | 0.63 | 0.57 |
| HyperSED | **0.80** | 0.89 | 0.82 | **0.79** | **0.85** | **0.80** | **0.78** |
| Proposed | 0.75 | **0.89** | **0.83** | 0.78 | 0.82 | 0.79 | **0.78** |

Table 8: NMI: Proposed vs. Unsupervised on English data.

|  | $B_1$ | $B_2$ | $B_3$ | $B_4$ | $B_5$ | $B_6$ | $B_7$ |
|---|---|---|---|---|---|---|---|
| HISEvent | 0.40 | 0.79 | 0.86 | 0.77 | 0.81 | 0.74 | 0.59 |
| HyperSED | **0.84** | 0.84 | 0.85 | 0.77 | 0.84 | 0.86 | **0.85** |
| Proposed | 0.81 | **0.86** | **0.91** | **0.79** | **0.88** | **0.93** | 0.75 |
|  | $B_8$ | $B_9$ | $B_{10}$ | $B_{11}$ | $B_{12}$ | $B_{13}$ | $B_{14}$ |
| HISEvent | 0.64 | 0.79 | 0.76 | 0.78 | **0.84** | 0.77 | 0.80 |
| HyperSED | 0.87 | 0.86 | 0.86 | 0.86 | 0.78 | **0.82** | 0.84 |
| Proposed | **0.88** | **0.87** | **0.90** | **0.87** | 0.75 | **0.82** | **0.89** |
|  | $B_{15}$ | $B_{16}$ | $B_{17}$ | $B_{18}$ | $B_{19}$ | $B_{20}$ | $B_{21}$ |
| HISEvent | 0.66 | 0.76 | 0.80 | 0.79 | 0.83 | 0.66 | 0.59 |
| HyperSED | **0.81** | **0.90** | 0.83 | **0.80** | **0.86** | **0.83** | **0.80** |
| Proposed | 0.77 | **0.90** | **0.84** | 0.79 | 0.83 | 0.82 | 0.79 |

Table 9: ARI: Proposed vs. Unsupervised on French data.

|  | $B_1$ | $B_2$ | $B_3$ | $B_4$ | $B_5$ | $B_6$ | $B_7$ | $B_8$ |
|---|---|---|---|---|---|---|---|---|
| HISEvent | 0.55 | 0.61 | 0.49 | 0.47 | 0.51 | 0.61 | 0.62 | 0.79 |
| HyperSED | **0.89** | 0.77 | 0.81 | 0.59 | 0.57 | 0.86 | 0.75 | 0.64 |
| Proposed | **0.89** | **0.94** | **0.92** | **0.90** | **0.86** | **0.93** | **0.93** | **0.89** |
|  | $B_9$ | $B_{10}$ | $B_{11}$ | $B_{12}$ | $B_{13}$ | $B_{14}$ | $B_{15}$ | |
| HISEvent | 0.43 | 0.53 | 0.56 | 0.77 | 0.74 | 0.78 | 0.69 | |
| HyperSED | 0.42 | 0.64 | 0.77 | 0.72 | 0.79 | 0.77 | 0.74 | |
| Proposed | **0.78** | **0.87** | **0.90** | **0.85** | **0.95** | **0.92** | **0.89** | |

Table 10: AMI: Proposed vs. Unsupervised on French data.

|  | $B_1$ | $B_2$ | $B_3$ | $B_4$ | $B_5$ | $B_6$ | $B_7$ | $B_8$ |
|---|---|---|---|---|---|---|---|---|
| HISEvent | **0.77** | 0.77 | 0.75 | 0.71 | 0.75 | 0.80 | 0.80 | **0.85** |
| HyperSED | **0.77** | 0.75 | 0.71 | 0.70 | 0.65 | 0.77 | 0.66 | 0.78 |
| Proposed | 0.76 | **0.84** | **0.82** | **0.79** | **0.77** | **0.84** | **0.83** | 0.79 |
|  | $B_9$ | $B_{10}$ | $B_{11}$ | $B_{12}$ | $B_{13}$ | $B_{14}$ | $B_{15}$ | |
| HISEvent | **0.71** | **0.79** | **0.82** | **0.84** | 0.85 | **0.87** | **0.82** | |
| HyperSED | 0.62 | 0.71 | 0.75 | 0.81 | 0.81 | 0.76 | 0.75 | |
| Proposed | 0.62 | 0.75 | 0.78 | 0.81 | **0.87** | 0.85 | 0.80 | |

Table 11: NMI: Proposed vs. Unsupervised on French data.

|  | $B_1$ | $B_2$ | $B_3$ | $B_4$ | $B_5$ | $B_6$ | $B_7$ | $B_8$ |
|---|---|---|---|---|---|---|---|---|
| HISEvent | **0.78** | 0.77 | 0.75 | 0.72 | 0.77 | 0.81 | 0.8 | **0.86** |
| HyperSED | 0.77 | 0.75 | 0.71 | 0.70 | 0.68 | 0.78 | 0.67 | 0.79 |
| Proposed | 0.77 | **0.84** | **0.82** | **0.79** | **0.78** | **0.85** | **0.84** | 0.80 |
|  | $B_9$ | $B_{10}$ | $B_{11}$ | $B_{12}$ | $B_{13}$ | $B_{14}$ | $B_{15}$ | |
| HISEvent | **0.72** | **0.80** | **0.83** | **0.85** | 0.86 | **0.88** | **0.83** | |
| HyperSED | 0.64 | 0.73 | 0.76 | 0.83 | 0.82 | 0.79 | 0.76 | |
| Proposed | 0.65 | 0.77 | 0.79 | 0.82 | **0.87** | 0.86 | 0.81 | |

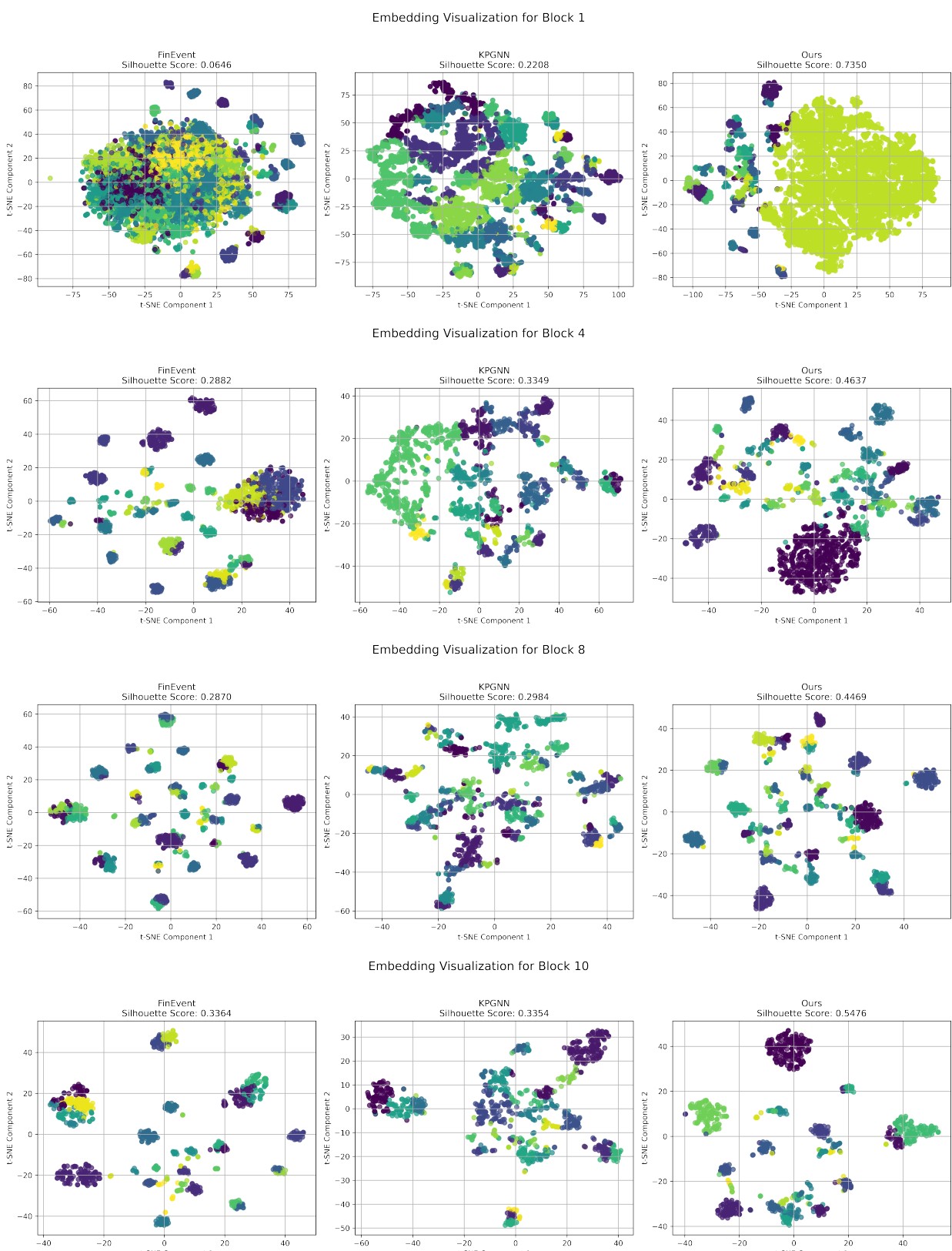

Figure 13: A comparison of embedding visualisations across different blocks. FinEvent achieves a higher NMI than KPGNN by correctly identifying the underlying event classes, while its lower silhouette score is attributed to the non-convex and overlapping nature of these true clusters, which penalises the distance-based metric. Also, FinEvents focus on optimising for NMI results in geometrically complex clusters are penalised by the distance-based silhouette score. Our affinity and alignment loss helps in clearer cluster separation, indicated by our higher silhouette score as well as higher values across the metrics.

Table 12: Tweets in the Predicted Cluster 11 for the block $B_{15}$

| Node ID | Tweet Text | Date | Ground Truth Label | Predicted Cluster |
|---|---|---|---|---|
| 1419 | Full match share it with your friends Hamid rahimi kabul 2012 http://t.co/zWJd5D0F | 2012-10-31 01:52:27 | 379 | 11 |
| 1428 | http://t.co/kcvxjLQA – Iran: Further information: human rights lawyer on hunger strike: Nasrin Sotoudeh,#IamNasrin,#FreeNasrin | 2012-10-31 02:25:17 | 253 | 11 |
| 1516 | Another Somali journalist killed 18th this year | 2012-10-31 04:54:13 | 370 | 11 |
| 1531 | Congratulation Mr Hamid Rahimi. We are proud of you. http://t.co/oKVClNjo | 2012-10-31 05:02:43 | 379 | 11 |
| 1713 | Hamid Rahimi on the way to training in Hamburg short interview http://t.co/aQlz5t38 | 2012-10-31 08:52:29 | 379 | 11 |
| 1764 | Alan Bennett laments nation turned into 'captive market' where public life exhibits a 'diminution of magnanimity' http://t.co/IAbyDCYs | 2012-10-31 10:11:43 | 373 | 11 |
| 1776 | Somali poet Warsame Shire Awale killed by gunmen: 30 October 2012 Last updated at 10:27 GMT Warsame Shire Awale ... http://t.co/0s1R7Z6f | 2012-10-31 10:31:21 | 370 | 11 |
| 1812 | Gunmen kill famous poet in Mogadishu http://t.co/jdNroA3l | 2012-10-31 11:35:05 | 370 | 11 |
| 1819 | please pray all the casualties in US-hurricane sandy | 2012-10-31 11:47:07 | 69 | 11 |
| 1946 | Somali poet Warsame Shire Awale killed: MOGADISHU, Somalia, Oct. 31 (UPI) – No arrests have been made in this ... http://t.co/TJtvooJB | 2012-10-31 14:43:23 | 370 | 11 |
| 2015 | #LoveMyL 23 die at Saudi Arabia wedding after celebratory gunfire downs electric cable - http://t.co/ucqcwUNg (bl... http://t.co/9rVlJcJP | 2012-10-31 17:27:38 | 435 | 11 |
| 2096 | Watching Man U Chelsea Via TL. Ofcourse. | 2012-10-31 20:00:37 | 54 | 11 |
| 2139 | Giggs! Good shoot old man! 0-1 | 2012-10-31 20:08:14 | 54 | 11 |
| 2314 | HT: Liverpool 0-1 Swansea City | Chico '34 #futsalsemut | 2012-10-31 21:05:31 | 160 | 11 |
| 2349 | Liverpool 1-2 Swansea. Norwich 2-1 Spurs. | 2012-10-31 21:37:26 | 160 | 11 |
| 2351 | Liverpool are up their old trick too"@AdoreLadies: Somewhere else in England, spurs are gettin embarrassed by Norwich..." | 2012-10-31 21:38:20 | 160 | 11 |
| 2364 | She work hard for her money | 2012-10-31 21:53:19 | 378 | 11 |
| 2368 | Liverpool 1-3 Swansea. No joke required. | 2012-10-31 21:55:35 | 160 | 11 |
| 2372 | Full time, Liverpool 1-3 Swansea City | 2012-10-31 21:55:58 | 160 | 11 |
| 2435 | I really want to see The Stone Roses omg anyone want to come with me to London in June??? | 2012-10-31 22:16:53 | 375 | 11 |
| 2452 | chelsea shoots 23 times. Amazing | 2012-10-31 22:19:20 | 54 | 11 |
| 2475 | so many kids coming to my house for candy, why cant i #trickortreat | 2012-10-31 22:27:56 | 150 | 11 |
| 2538 | Alan Bennett rips into the culture industry subtly and hilariously in People. I'll never look at a stately home in the same way again | 2012-10-31 23:08:48 | 373 | 11 |

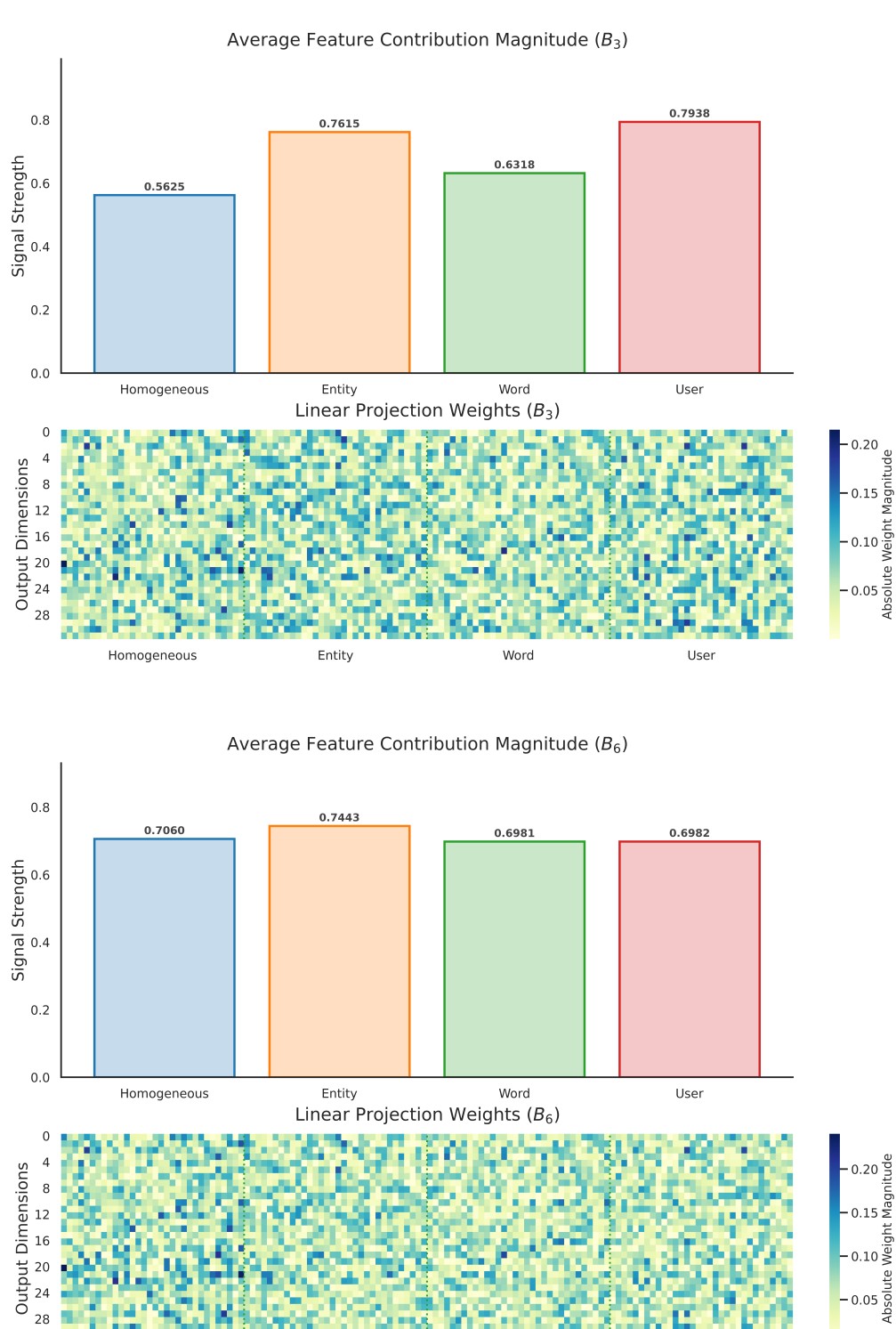

Figure 14: Layer-wise Feature Contribution and Weight Analysis for Blocks $B_3$ and $B_6$ for the English dataset. The bar charts illustrates the average signal strength of each relation type (Homogeneous, Entity, Word, User), calculated as the mean L2 norm of the contribution matrix (input features projected by learned weights). The bottom panel visualizes the absolute magnitude of the linear projection weights (W), where the x-axis corresponds to the concatenated input features. Vertical dotted lines demarcate the input sections specific to each relation type.

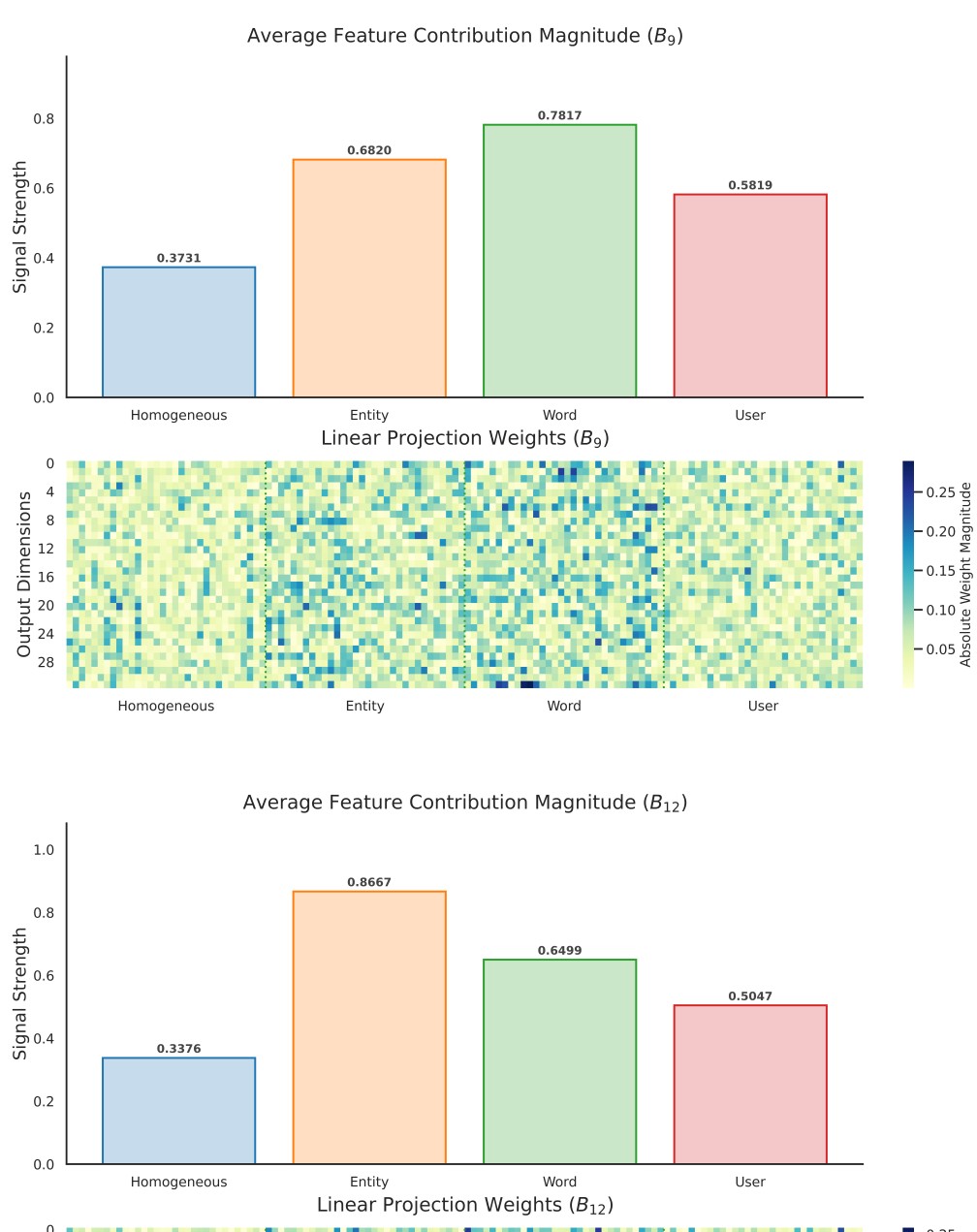

Figure 15: Layer-wise Feature Contribution and Weight Analysis for Blocks $B_9$ and $B_{12}$ for the French dataset. The bar charts illustrates the average signal strength of each relation type (Homogeneous, Entity, Word, User), calculated as the mean L2 norm of the contribution matrix (input features projected by learned weights). The bottom panel visualizes the absolute magnitude of the linear projection weights (W), where the x-axis corresponds to the concatenated input features. Vertical dotted lines demarcate the input sections specific to each relation type.

---

**Algorithm 1** Incremental Block-Wise Learning (Latest Message Strategy)

---

**Require:** Stream $S$; Window $W$; Epochs $E$; Encoder $\mathcal{E}_\theta$;
**Ensure:** Event clusters for each time block
  1: **Data Preprocessing:**
  2: Split $S$ into $T$ temporal blocks $\{B_0, \ldots, B_T\}$.
  3: Initialize parameters $\theta$ randomly.
     **Phase 1: Initial Training (Block 0)**
  4: Construct Graph $H_0$ using messages in $B_0$.
  5: Extract matrices $A_{hom}^{(0)}, A_{ent}^{(0)}, A_{word}^{(0)}, A_{user}^{(0)}$.
  6: **for** epoch $e = 1$ to $E$ **do**
  7:     $Z_0 \leftarrow \mathcal{E}_\theta(A_*^{(0)}, X_0)$
  8:     $\mathcal{L}_{final} \leftarrow \mathcal{L}_{trip} + \mathcal{L}_{affinity} + \mathcal{L}_{align}$
  9:     Update $\theta \leftarrow \theta - \eta \nabla_\theta \mathcal{L}_{final}$
 10: **end for**
     **Phase 2: Incremental Inference**
 11: **for** block index $t = 1$ to $T$ **do**
     **Graph Construction:**
 12:     Construct Graph $H_t$ using messages in $B_t$.
         *// Nodes/edges from $B_{0 \ldots t-1}$ are removed; knowledge stays in $\theta$.*
 13:     Generate Adjacency Matrices $A_*^{(t)}$ for block $t$.
     **Inference:**
 14:     $Z_t \leftarrow \mathcal{E}_\theta(A_*^{(t)}, X_t)$ using current $\theta$.
 15:     Cluster $Z_t$ to detect event sets $C_t$.
 16:     Report metrics (NMI, AMI, ARI).
     **Maintenance (Conditional Training):**
 17:     **if** $t \mod W == 0$ **then**
 18:         **for** epoch $e = 1$ to $E$ **do**
 19:             $Z_t \leftarrow \mathcal{E}_\theta(A_*^{(t)}, X_t)$
 20:             Calculate $\mathcal{L}_{final}$.
 21:             Update $\theta \leftarrow \theta - \eta \nabla_\theta \mathcal{L}_{final}$
 22:         **end for**
 23:     **end if**
 24: **end for**

---

---

**Algorithm 2** Multi-head Cross-Layer Attention Module

---

1: **Input:** Combined layer features $\mathcal{X}_{G_c}$, Heterogeneous layer features $\{\mathcal{X}_{G_e}, \mathcal{X}_{G_w}, \mathcal{X}_{G_u}\}$, Number of heads $H$.
2: **Output:** Fused node features $\mathcal{X}_{attn}$.
3: **Parameters:** Learnable weight matrices $W_Q, W_K, W_V, W_{attn}$, bias $b_{attn}$.
   *// See Eq. 2 and Eq. 3 in main paper*
4: Set Query $Q \leftarrow \mathcal{X}_{G_c}$ {Homogeneous features act as Query}
5: Initialize list of attention outputs $L_{out} \leftarrow [\mathcal{X}_{G_c}]$
6: **for** each relation graph $r \in \{G_e, G_w, G_u\}$ **do**
7:     Set Key $K \leftarrow \mathcal{X}_r$, Value $V \leftarrow \mathcal{X}_r$
8:     Compute Attention Score: $\alpha_r = \text{softmax}\left(\frac{(QW_Q)(KW_K)^T}{\sqrt{d}}\right)$
9:     Compute Head Output: $\mathcal{X}_r^{out} = \alpha_r(VW_V)$
10:     Append $\mathcal{X}_r^{out}$ to $L_{out}$
11: **end for**
   *// Concatenate features from Combined layer and all Heterogeneous attention outputs*
12: $H_{concat} \leftarrow \text{Concat}(L_{out})$
   *// Final Linear Projection (Eq. 6)*
13: $\mathcal{X}_{attn} \leftarrow H_{concat}W_{attn}^T + b_{attn}$
14: **return** $\mathcal{X}_{attn}$

---

**Algorithm 3** Joint Loss Optimization (Affinity & Alignment)

---

1: **Input:** Batch features $Z$ (from Encoder), Batch labels $Y$, Margins $m, m_1, m_2$, Temperature $\sigma$.
2: **Output:** Total Loss $\mathcal{L}_{final}$.
3: **Networks:** Label Encoder $\mathcal{E}_y$, Feature Projector $\mathcal{E}_x$.
   *Triplet Loss Calculation*
4: Sample triplets $(a, p, n)$ from $Z$ using semi-hard/hard mining.
5: $\mathcal{L}_{trip} \leftarrow \max(0, ||Z_a - Z_p||^2 - ||Z_a - Z_n||^2 + m)$
   *Affinity Loss Calculation (Eq. 7-8)*
6: Compute Feature Similarity Matrix (RBF Kernel):
7: $S_{feat}[i,j] \leftarrow \exp\left(-\frac{||Z_i - Z_j||^2}{\sigma}\right)$
8: Compute Label Similarity Matrix with $m_1$ and $m_2$:
9: $S_{label}[i,j] \leftarrow \mathbb{1}(Y_i == Y_j) \cdot m_1 + (1 - \mathbb{1}(Y_i == Y_j)) \cdot m_2$
10: Mask diagonal elements to 0.
11: $\mathcal{L}_{affinity} \leftarrow ||S_{feat} - S_{label}||_F$
   *Alignment Loss Calculation (Eq. 9-13)*
12: Project features: $E_x \leftarrow \mathcal{E}_x(Z)$
13: Project labels: $E_y \leftarrow \mathcal{E}_y(Y)$
14: Compute Cosine Similarity:
15: $CosSim(i) \leftarrow \frac{E_x[i] \cdot E_y[i]}{||E_x[i]||_2 \cdot ||E_y[i]||_2}$
16: $\mathcal{L}_{align} \leftarrow 1 - \frac{1}{N}\sum_{i=1}^{N} CosSim(i)$
   *Total Loss*
17: $\mathcal{L}_{final} \leftarrow \mathcal{L}_{trip} + \mathcal{L}_{affinity} + \mathcal{L}_{align}$
18: **return** $\mathcal{L}_{final}$

---