# OpenReview forum: "Enhancing Deep Consistent Graph Metric with Affinity and Alignment for Incremental Social Event Detection using Cross-Layer Attention"
_TMLR — Accepted by TMLR_

### Review · Reviewer_dgQK · 2025-11-23

**Summary Of Contributions:**

This paper takes social event detection beyond triplet loss by enforcing both local (within-batch) and global (across-batch) consistency via new regularized losses and scalable attention, achieving state-of-the-art results on real-world datasets.

**Audience:**

Yes

**Audience Explanation:**

Yes, at least some individuals in TMLR's audience would be interested in the findings of this paper, as it addresses key limitations of deep metric learning—such as latent space consistency and scalable attention mechanisms—which are central topics in machine learning research. While the application is focused on social event detection, the proposed affinity and alignment loss functions and the cross-layer attention mechanism generalize to other areas of deep metric learning, potentially offering methodological insights relevant for broader ML problems. Thus, its findings hold value for researchers studying representation learning, graph neural networks, and scalable learning algorithms in practical settings.

**Claims And Evidence:**

Yes

**Claims Explanation:**

The submission provides extensive analysis comparing the proposed approach to supervised and unsupervised baselines, featuring both quantitative and qualitative evaluations across multiple metrics (NMI, AMI, ARI) on two datasets (English and French). Statistical significance is established using the Wilcoxon signed-rank test, confirming that gains over the best existing methods are highly significant. The ablation study further demonstrates that each model component—the new attention module, affinity loss, and alignment loss—contributes distinct, measurable improvements. Some minor limitations are acknowledged: the method may occasionally merge closely related consecutive events, impacting certain metrics (NMI/AMI) but not undermining the overall advantage. Thus, the experimental design, theoretical support, and comprehensive empirical results together provide clear, consistent, and convincing evidence for the claimed contributions and superiority of this work.

**Requested Changes:**

Critical Adjustments (necessary for acceptance):

- Clarity and Limitations: Explicitly discuss and quantify cases where closely related consecutive events are merged by the model, affecting NMI/AMI scores. Suggest mitigation strategies or clarify when this behavior is or is not desirable.

- Generalizability: Strengthen discussion about how the proposed affinity and alignment loss functions can be applied to other domains beyond social event detection, especially for TMLR's machine learning-focused audience.

Non-Critical Adjustments (would strengthen the work):

- Implementation Details: Increase transparency regarding data splits, hyperparameters, and model settings to foster reproducibility.

- Related works: Zhang Y, Cao D, Liu Y. Counterfactual neural temporal point process for estimating causal influence of misinformation on social media[J]. Advances in Neural Information Processing Systems, 2022, 35: 10643-10655.

- Supplementary Materials: Add sample code or pseudo-code for the proposed losses and attention mechanisms.

- Visualization: Include more visual examples showcasing model behaviours (e.g., alignment in embedding space, event clusters).

---

> ### Author Response · Authors · 2025-12-11
> **Response to the Reviewer**
>
> We appreciate the thoughtful comments of the reviewer and appreciating our paper ''extensive analysis comparing the proposed approach to supervised and unsupervised baselines, featuring both quantitative and qualitative evaluations...''
>
> Here we address some of the Critical adjustments suggested by the reviewer:
>
> 	Q1. Clarity and Limitations: Explicitly discuss and quantify cases where closely related consecutive events are merged by the model, affecting NMI/AMI scores. Suggest mitigation strategies or clarify when this behavior is or is not desirable.
>
> Based on the suggestions of the reviewer we have added an ablation to highlight when and why events are merged in our model in Ablation Study -> Failure Case Analysis (Pages 11, 12).
>
> 	Q2.   Generalizability: Strengthen discussion about how the proposed affinity and alignment loss functions can be applied to other domains beyond social event detection, especially for TMLR's machine learning-focused audience.
>
> We have shown the generalizability of our proposed loss in "Ablation Study -> Generalizability of the Proposed Loss" (Page 11) where we compare the class-wise feature separation of our loss to 25 other metric learning losses across various graph datasets.
>
> 	Q3. Implementation Details: Increase transparency regarding data splits, hyperparameters, and model settings to foster reproducibility.
>
> We have added detailed implementation in the appendix (Pages 19 and 20).
>
> 	Q4. Related works: Zhang Y, Cao D, Liu Y. Counterfactual neural temporal point process for estimating causal influence of misinformation on social media[J]. Advances in Neural Information Processing Systems, 2022, 35: 10643-10655.
>
> Thank you for the suggestion of the excellent work on misinformation, we have added the citation in the paper.
>
> 	Q5. Supplementary Materials: Add sample code or pseudo-code for the proposed losses and attention mechanisms.
>
> We have added detailed pseudocodes (Algorithm 1 , 2 and 3) in the supplementary in Page 29.
>
> 	Q6. Visualization: Include more visual examples showcasing model behaviours (e.g., alignment in embedding space, event clusters).
>
> We have added more visualization in Figures 8 and 9 of the main paper and Figures 14 and 15 in the supplementary.
>
> Kindly note: All the changes in the updated manuscript are highlighted in blue for the ease of reviewing.

---

> ### Author Response · Authors · 2025-12-15
> **Response to the Reviewer**
>
> We would like to thank the reviewer again for the valuable feedback. As the discussion period draws to a close tomorrow, we wanted to reach out to ensure that our previous responses have adequately addressed your concerns. We remain fully available to clarify any remaining points or answer additional questions you might have.

---

### Review · Reviewer_xQJE · 2025-11-24

**Summary Of Contributions:**

To address the limitations of existing methods in event detection, particularly those based on triplet loss, the authors propose a hierarchical modeling approach and two novel loss functions. Experimental results demonstrate a significant performance improvement.

**Strengths**
1. The authors' motivation and approach are well-motivated, straightforward, and clear, pinpointing the key limitation of the triplet loss.
2. The method proposed by the authors achieves highly competitive results, with the visualization in Fig. 10 being particularly remarkable.

**Weakness**
1. Although the proposed two key losses demonstrate superior performance in the experimental results (Fig. 7), they do not appear to be specific to the problem of Event Detection. In other words, these loss strategies may be adapted to arbitrary classification tasks. They do not show what makes Event Detection different or difficult.
2. I am uncertain about how the authors set up the label information, i.e., $f_y(y_i)$ . To my knowledge, in scenarios like continual learning where new categories (which would correspond to new event types in this work) are introduced, it is crucial to explain how the model maintains consistent dimensions for label representation or how these new event types are defined.  Lemmas 6 and 7 suffer from similar issues.
3. Some of the lemmas appear unduly cumbersome. For example, the detailed exposition of foundational concepts like the triangle inequality in Lemma 4 seems dispensable in the main text and could be moved to the Appendix. The authors should focus on providing rigorous reasons for why these losses satisfies the conditions of the Discriminative Graph.
4. Other typos. In Eq. 1, $\sum_{\forall s_n \in V}$ appears to be incorrect. It should probably be over the neighbour nodes, i.e., $\forall s_n \in N_{s_n}$？

In summary, my primary concerns are the oversimplified loss configuration and some lemmas are too redundant.


**Update**
The authors have addressed the majority of my concerns.

**Audience:**

Yes

**Audience Explanation:**

The hierarchical processing proposed by the authors is relatively novel. However, the two proposed loss functions are somewhat conventional, as they are not specifically tailored for the event detection scenario.

**Claims And Evidence:**

Yes

**Claims Explanation:**

The method proposed by the authors is effective, as clearly demonstrated by the significant results. Furthermore, ablation studies and visualizations also validate the efficacy of the proposed modules.

**Requested Changes:**

Please refer to the weaknesses.

---

> ### Author Response · Authors · 2025-12-11
> **Response to the Reviewer**
>
> We appreciate the detailed comments about our paper and thank the reviewer for acknowledging our "visualization" and regarding our work as "well-motivated, straightforward, and clear...".
>
> Here are our comments on the weaknesses highlighted by the reviewer and the action taken to resolve the same.
>
> 	Q1. Although the proposed two key losses demonstrate superior performance in the experimental results (Fig. 7), they do not appear to be specific to the problem of Event Detection. In other words, these loss strategies may be adapted to arbitrary classification tasks. They do not show what makes Event Detection different or difficult.
>
> We agree that the proposed losses can be applied to broader domains. Specific to the event detection (or similar) problem, we proposed cross-layer attention where heterogeneous interactions between incoming social messages prevails. Existing metric learning methods do not address this issue. In other words, the proposed losses alone (without cross-layer attention) do not solve all the issues related to event detection problem rather it reduces the issues generated from the triplet loss.
>
> As our proposed loss strategy can be used independently or as a regularizer to other metric learning losses across diverse tasks, we report a new generalizability study in section "Ablation Study -> Generalizability of the Proposed Loss" (main paper). We compare the class-wise feature separation of our loss to other 25 different metric learning losses across various graph datasets.
>
> 	Q2. I am uncertain about how the authors set up the label information, i.e., $f_y(y_i)$. To my knowledge, in scenarios like continual learning where new categories (which would correspond to new event types in this work) are introduced, it is crucial to explain how the model maintains consistent dimensions for label representation or how these new event types are defined. Lemmas 6 and 7 suffer from similar issues.
>
> We use integer labels for $y_i$ and map them to a $d$-dimensional space using 2-layer MLP ($f_y$). A new event class when appear during incremental-training phase is assigned with a new integer number and mapped to the same label space by $f_y$ keeping the projections of old event classes. For both initial training and incremental-training, the projection vector (label embedding) dimensions are kept the same.
>
> We could not able to understand the issues related to Lemma 6 and 7. To clarify, the dimensions of $L'$ and $L''$ (Lemma 5) are same as both represents the label embedding of different mini-batches from the same block rather than different blocks. Please let us know issue in detail so that we can clarify the same in more details.
>
>
> 	Q3. Some of the lemmas appear unduly cumbersome. For example, the detailed exposition of foundational concepts like the triangle inequality in Lemma 4 seems dispensable in the main text and could be moved to the Appendix. The authors should focus on providing rigorous reasons for why these losses satisfies the conditions of the Discriminative Graph.
>
> Based on the reviewers suggestion we have moved the proof of Lemma 4, to the Appendix to improve readability.
>
> For the other point: A discriminative Graph is defined as satisfying $\alpha_c < \beta_c$ where $\alpha_c$ is the maximum intra-event distance and $\beta_c$ is the smallest inter-event distance. Our proposed losses directly enforce these constraints. The *Affinity Loss* minimizes the Frobenius norm difference between the feature adjacency matrix $S_{feature}$ and the ideal label matrix $S_{label}$; for intra-event pairs where $S_{label}=1$, this forces $S_{feature} \to 1$ and consequently the feature distance $\|x_i - x_j\|^2 \to 0$ (minimizing $\alpha_c$), while for inter-event pairs where $S_{label}=0$, it forces $S_{feature} \to 0$, maximizing the distance (increasing $\beta_c$) . In this way we can fix the loss to specific intra-class ($\alpha_c$) and inter-class ($\beta_c$) distance by specifying parameters $m_1$ and $m_2$ such that $\beta_c>\alpha_c$, thus making the loss  satisfy the property of a discriminative graph. Our ablations shows that $m_1=1, m_2=0$ works well in practice. Furthermore, the Alignment Loss addresses the issue of batch inconsistency by anchoring the feature space to a fixed, learnable label space via cosine similarity, ensuring that features from the same event across different mini-batches align with the same label embedding, thereby satisfying the transitivity required for a global discriminative graph structure. These theoretical mechanisms are explicitly realized in our implementation, to enforce these constraints during training.
>
> 	Q4. Other typos. In Eq. 1, appears to be incorrect. It should probably be over the neighbour nodes.
>
> We are thankful to the reviewer for pointing this mistake. We have corrected it in the updated manuscript.
>
> Kindly note: All the changes in the updated manuscript are highlighted in blue for the ease of reviewing.

---

> ### Author Response · Authors · 2025-12-15
> **Response to the Reviewer**
>
> We would like to thank the reviewer again for the valuable feedback. As the discussion period draws to a close tomorrow, we wanted to reach out to ensure that our previous responses have adequately addressed your concerns. We remain fully available to clarify any remaining points or answer additional questions you might have.

---

> ### Comment · Reviewer_xQJE · 2025-12-15
> **Summary**
>
> I apologize for the delayed response due to personal reasons. I have revisited the authors’ rebuttal and carefully reviewed the comments from the other reviewers. Most of my concerns have been addressed. It is worth noting that the authors have resolved Q2, and as Lemmas 6 and 7 represent the same issue, no further explanation is required. I still have a mild concern regarding Q1, even though the authors have further examined the generalization of the proposed loss.  Therefore, I lean toward weak accept. If the other reviewers support the acceptance of this paper, I would be willing to support their position.

---

### Review · Reviewer_SV4W · 2025-12-01

**Summary Of Contributions:**

This paper introduces a new deep metric learning framework for incremental social event detection, addressing the fundamental weaknesses of triplet-loss-based models—namely inconsistent intra-/inter-event distances and the inability to align same-event samples across mini-batches. We propose two novel regularizers, Affinity Loss and Alignment Loss, which jointly enforce discriminative structure by increasing intra-event affinity and aligning features to a shared label space, theoretically matching the discriminative guarantees of CGML while avoiding its expensive pairwise sampling. Alongside a cross-layer multi-head attention module that effectively integrates heterogeneous relations without extra optimization, the model achieves substantial gains on Event2012 and Event2018, improving over supervised SOTA baselines by 26.59% (NMI), 30.49% (AMI), and 142.38% (ARI) on average, and outperforming unsupervised methods as well. The framework scales linearly with mini-batches, supports efficient incremental training, and offers a broadly applicable solution for metric learning tasks suffering from feature inconsistency.

**Audience:**

Yes

**Audience Explanation:**

1. The paper proposes new loss functions (Affinity & Alignment) with solid theoretical backing, directly addressing well-known weaknesses of triplet loss. This contributes to the broader research agenda on robust metric learning, discriminative representations, and training stability—topics central to TMLR’s audience.
2. Incremental social event detection is an active and socially important problem involving streaming data, heterogeneous graphs, and online learning. Readers interested in real-world applications of representation learning—especially in social media analysis, crisis detection, and temporal graph mining—will find the approach directly useful.
3. The proposed Affinity/Alignment losses and cross-layer attention module are generic and can be plugged into many other graph-based or minibatch-based learning systems. TMLR readers who work on GNNs, heterogeneous data, or scalable learning algorithms may find these components directly applicable to their own domains.

**Claims And Evidence:**

Yes

**Claims Explanation:**

1. The paper provides formal proofs and lemmas demonstrating that the proposed Affinity and Alignment losses achieve discriminative properties comparable to CGML, but with far lower computational cost. This theoretical grounding directly supports the claim that the new losses produce consistent latent-space structure.
2. Across two public large-scale datasets (Event2012 and Event2018), the method consistently outperforms all supervised and unsupervised baselines by large margins—e.g., NMI/AMI/ARI improvements of 26.59%, 30.49%, and 142.38%, respectively—providing clear, quantitative evidence that the proposed approach is effective in practice.
3. The paper goes beyond raw scores by including Wilcoxon signed-rank tests showing the improvements are statistically significant (p < 0.0001 in most cases). Detailed ablation studies further isolate the contributions of each component (attention module, Affinity Loss, Alignment Loss), confirming that every claim is causally supported by controlled experiments.

**Requested Changes:**

1. The paper claims broad applicability of Affinity/Alignment losses beyond event detection, but the experiments are limited to two datasets from similar domains. Clarify the generality claims and, if possible, provide a small supplemental experiment or discussion showing applicability to other metric-learning or graph-learning settings.
2. Although classical baselines are included, comparisons with more recent deep metric learning methods are not sufficiently explored. Add justification for baseline selection or include one additional modern metric-learning baseline to strengthen the empirical comparison.
3. The ablation only evaluates the presence or absence of modules, but does not analyze sensitivity to key hyperparameters. Include a short sensitivity analysis or discussion explaining hyperparameter stability.
4. The paper briefly mentions performance drops on closely related consecutive events, but the explanation is superficial. Provide clearer examples or a deeper explanation of when alignment may cause cluster merging, and propose possible remedies.
5. The role of each heterogeneous relation (words, entities, users) is not quantified, making it hard to assess the value of the cross-layer attention. Add a relation-importance analysis or a brief visualization demonstrating how the attention weights differ across relations.
6. The description of block-wise incremental learning and the “latest message” strategy lacks low-level details. Include a more explicit description or pseudo-code of the incremental training pipeline to improve reproducibility.

---

> ### Author Response · Authors · 2025-12-11
> **Response to the Reviewer**
>
> We appreciate and thank the detailed comments about our paper. This helped us in improving our paper further. Also, we thank the reviewer for the appreciation the "theoretical grounding" and other aspects of our work.
>
> To further address the reviewers concerns, here are our responses on the weakness and action taken to resolve the same.
>
> 	Q1. The paper claims broad applicability of Affinity/Alignment losses beyond event detection, but the experiments are limited to two datasets from similar domains. Clarify the generality claims and, if possible, provide a small supplemental experiment or discussion showing applicability to other metric-learning or graph-learning settings.
>
> We now add a new experiment on generalizability in section, "Generalizability of the Proposed Loss," on Page 11. We evaluated our proposed Affinity and Alignment losses against 25 different metric learning losses on four standard benchmark datasets: Cora (citation), Amazon Computers/Photo (co-purchase), and Flickr (social). The results, presented in Table 3 (Page 11), show that our method consistently matches or outperforms state-of-the-art baselines in the general clustering task.
>
> 	Q2. Although classical baselines are included, comparisons with more recent deep metric learning methods are not sufficiently explored. Add justification for baseline selection or include one additional modern metric-learning baseline to strengthen the empirical comparison.
>
> We have added a deep metric learning based baseline in the Tables 2 (main paper), 4 and 5 (supplementary) as suggested by the reviewer. We choose this baseline as it is not only one of the recent works in this field but also because it optimizes for semantic consistency within neighborhoods, allowing it to better group diverse instances of the same event while ignoring irrelevant noises from heterogeneous connections.
>
> 	Q3. The ablation only evaluates the presence or absence of modules, but does not analyze sensitivity to key hyperparameters. Include a short sensitivity analysis or discussion explaining hyperparameter stability.
>
> We have added a detailed sensitivity analysis for the temperature parameter ($\sigma$) and the margin parameters ($m_1​,m_2$​). The discussion is provided in the "Additional results" section on Page 20. The corresponding results are visualized in Figures 11 and 12 (Page 21) of the Appendix. The analysis indicates that $\sigma<1.0$ degrades performance, and the model performs best when the intra-class margin ($m_1$​) is distinct from the inter-class margin ($m_2$​), specifically at $m_1​=1,m_2​=0$.
>
> 	Q4. The paper briefly mentions performance drops on closely related consecutive events, but the explanation is superficial. Provide clearer examples or a deeper explanation of when alignment may cause cluster merging, and propose possible remedies.
>
> We provide a deeper analysis of cluster merging in the "Failure Case Analysis" section on Page 11. We specifically analyzed Block 15 (Cluster 11) of the English dataset, as visualized in Figure 8 and Figure 9 (Page 12). We select this Block as we obtain low NMI values here and Cluster 11 has the highest number of diverse labels in this block. The analysis reveals that the merging of distinct events is caused by noisy connections based on common entities and the concatenation of timestamp embeddings, which increases feature similarity between different events occurring in the same time window.
>
> 	Q5. The role of each heterogeneous relation (words, entities, users) is not quantified, making it hard to assess the value of the cross-layer attention. Add a relation-importance analysis or a brief visualization demonstrating how the attention weights differ across relations.
>
> To assess the value of the cross-layer attention, we have added a "Layer-wise Feature Contribution and Weight Analysis" in the Appendix. Figure 14 (Page 27) and Figure 15 (Page 28) visualize the average signal strength and linear projection weights for Homogeneous, Entity, Word, and User relations across different blocks for both datasets. This quantification demonstrates how the attention module dynamically weighs different heterogeneous attributes.
>
> 	Q6. The description of block-wise incremental learning and the “latest message” strategy lacks low-level details. Include a more explicit description or pseudo-code of the incremental training pipeline to improve reproducibility.
>
> We have clarified the training pipeline in the "Incremental Block Wise Training" section on Page 7. Furthermore, for more clarity and also to ensure reproducibility, we have included the complete pseudo-code for the incremental training pipeline in Algorithm 1: "Incremental Block-Wise Learning (Latest Message Strategy)" on Page 29. This algorithm explicitly details the data preprocessing, initial training, and the loop for incremental inference and maintenance.
>
> Kindly note: All the changes in the updated manuscript are highlighted in blue for the ease of reviewing.

---

> ### Author Response · Authors · 2025-12-15
> **Response to the Reviewer**
>
> We would like to thank the reviewer again for the valuable feedback. As the discussion period draws to a close tomorrow, we wanted to reach out to ensure that our previous responses have adequately addressed your concerns. We remain fully available to clarify any remaining points or answer additional questions you might have.

---

### Comment · Action_Editor_Guzr · 2025-12-15
**Please acknowledge authors' responses**

Dear Reviewers,

Thank you for your contributions to the review process.

As the author discussion phase is nearing its end, I encourage you to carefully review the authors' responses and communicate with them directly if needed.

This interaction is essential for you to clarify any remaining questions or concerns and gather all necessary information before formulating your official recommendation with confidence.

Please do not hesitate to contact me if any points are unclear or require further clarification.

Best regards,
AE

---

### Decision · Action_Editor_Guzr · 2026-02-02

**Recommendation:** Accept with minor revision

**Additional Comments:**

Please specify the source for Figure 2 (I guess it is Event2012).

**Audience:**

Yes

**Audience Explanation:**

Event detection problems are a major topic in graph learning problems, which fall in the scope of TMLR. Therefore, the new methods in this field are of interest to TMLR’s audience who study graph learning problems.

**Claims And Evidence:**

Yes

**Claims Explanation:**

This paper proposes affinity loss and alignment loss, to solve the limitation of the triplet loss used to train graph-based models for event detection tasks.

This paper claims are as follows:
- The triplet loss suffers from two issues: (i) inconsistency in intra-event and inter-event
distances (ii) discrepancy between messages in the same event across different mini-batches.
- The proposed losses are introduced to solve the issues. Also, cross-layer attention is proposed to handle heterogeneous relations (e.g., common entities).

These claims are supported by empirical verifications on two publicly available datasets for social event detection tasks.